# Potential effect of the marine carbon cycle on the multiple equilibria window of the Atlantic Meridional Overturning Circulation

Amber A. Boot[1], Anna S. von der Heydt[1,2], and Henk A. Dijkstra[1,2]

[1]Institute for Marine and Atmospheric research Utrecht, Department of Physics,Utrecht University, Utrecht, the Netherlands
[2]Center for Complex Systems Studies, Utrecht University, Utrecht, the Netherlands

**Correspondence:** Amber A. Boot (a.a.boot@uu.nl)

**Abstract.** The Atlantic Meridional Overturning Circulation (AMOC) is considered to be a tipping element in the Earth System due to possible multiple (stable) equilibria. Here, we investigate the multiple equilibria window of the AMOC within a coupled ocean circulation-carbon cycle box model. We show that adding couplings between the ocean circulation and the carbon cycle model affects the multiple equilibria window of the AMOC. Increasing the total carbon content of the system widens the multiple equilibria window of the AMOC, since higher atmospheric $pCO_2$ values are accompanied by stronger freshwater forcing over the Atlantic Ocean. The important mechanisms behind the increase of the multiple equilibria window are the balance between the riverine source and the sediment sink of carbon and the sensitivity of the AMOC to freshwater forcing over the Atlantic Ocean. Our results suggest that changes in the marine carbon cycle can influence AMOC stability in future climates.

## 1 Introduction

The Atlantic Meridional Overturning Circulation (AMOC) plays a large role in modulating global climate (Vellinga and Wood, 2008; Palter, 2015) because it transports heat from the Southern to the Northern Hemisphere and is one of the prominent tipping elements in the Earth System (Lenton et al., 2008; Armstrong-McKay et al., 2022). Model studies suggest that the AMOC can have multiple stable equilibria: the on-state, representing the current AMOC state with a strong northward flow at the surface and a southward return flow at intermediate depths; and the off-state, representing a weak or even reversed AMOC state (Weijer et al., 2019). From a dynamical systems point of view, a bi-stable AMOC regime appears through the occurrence of two saddle-node bifurcations (Dijkstra, 2007) and the region in parameter space where both on- and off-states co-exist is the multiple equilibria window (MEW), also referred to as the bi-stability window (Barker and Knorr, 2021).

Climate variability in the past, such as Heinrich events, has been linked to tipping of the AMOC (Rahmstorf, 2002; Lynch-Stieglitz, 2017). Under anthropogenic forcing, the global warming threshold for AMOC tipping has been recently estimated to be around 4 °C (Armstrong-McKay et al., 2022). Recent studies suggest the AMOC has been weakening (Caesar et al., 2018; Dima et al., 2021), and might even collapse in this century (Ditlevsen and Ditlevsen, 2023). Using model data from the Coupled Model Intercomparison Project 6 (CMIP6, Eyring et al., 2016), a consistent weakening of the AMOC under future climate change is projected (Weijer et al., 2020), with a 34-45% decrease in AMOC strength in 2100, but no clear tipping was

found. However, these models may have a too stable AMOC (Weijer et al., 2019) affecting the probability of AMOC tipping before 2100. Under AMOC tipping, a strong cooling in the Northern Hemisphere (Rahmstorf, 2002; Drijfhout, 2015), changes in the water cycle (Vellinga and Wood, 2002; Jackson et al., 2015), and potential interactions with other tipping elements in the Earth System (Dekker et al., 2018; Wunderling et al., 2021; Sinet et al., 2023) are expected.

     The AMOC can also interact with the marine carbon cycle and therefore influence atmospheric $pCO_2$. By affecting the
transport of important tracers, such as dissolved inorganic carbon (DIC), total alkalinity, and nutrients, the AMOC affects the solubility and biological carbon pumps. Evidence for a coupling between the AMOC and marine carbon cycle is provided in proxy data (Bauska et al., 2021). Model studies show a wide range of potential carbon cycle responses to a collapse of the AMOC. While most models show an increase in atmospheric $pCO_2$ (e.g., Marchal et al., 1998; Schmittner and Galbraith, 2008; Matsumoto and Yokoyama, 2013; Boot et al., 2024), the magnitude and precise mechanisms are dependent on the model used
and climatic boundary conditions (Gottschalk et al., 2019).

     As the AMOC can influence atmospheric $pCO_2$, there is a potential feedback mechanism since atmospheric $pCO_2$ influences the hydrological cycle (Weijer et al., 2019; Barker and Knorr, 2021), which through changes in buoyancy fluxes, affects the AMOC. Previous studies, mostly focused on proxy data, suggest that there may be a relation between atmospheric $pCO_2$ and the MEW of the AMOC (Barker et al., 2010, 2015). However, a clear mechanistic view has not been given yet. Here, we study
the mechanisms of how the marine carbon cycle can affect the MEW of the AMOC using a coupled ocean circulation-carbon cycle box model.

## 2   Methods

We have coupled a box model suitable for simulating AMOC dynamics (Section 2.1) to a carbon cycle box model (Section 2.2). To be able to accurately represent atmospheric $CO_2$ concentrations, the coupled model extends the AMOC box model
by including boxes that represent the Indo-Pacific. Steady states of the coupled model, where several non-linear couplings are implemented (Section 2.3), are determined using continuation software (Section 2.4). Parameter values and model equations are described in Appendices B and C.

### 2.1   AMOC box model

The box model (Cimatoribus et al., 2014; Castellana et al., 2019) representing the AMOC dynamics simulates the depth of the
Atlantic Ocean pycnocline, and the distribution of salt in the Atlantic Ocean and the Southern Ocean. It consists of 5 boxes, with 6 prognostic variables. The northern box $n$ represents the regions of deep water formation in the North Atlantic and box $s$ represents the entire Southern Ocean (i.e. all longitudes). There are two thermocline boxes $t$ and $ts$ where box $ts$ represents the region between 30°S and 40°S which is characterized by strong sloping isopycnals where the pycnocline becomes shallower moving poleward. Underneath the four surface boxes, there is one box ($d$) representing the deep ocean.
The distribution of salinity in the boxes is dependent on the ocean circulation and surface freshwater fluxes. In the Southern Ocean, there is wind-induced Ekman transport into the Atlantic ($q_{Ek}$), and there is an eddy-induced transport from the Atlantic

into the Southern Ocean ($q_e$) which is dependent on the pycnocline depth D. The difference between the two, defined as $q_S = q_e$-$q_{Ek}$, represents upwelling in the Southern Ocean and net volume transport into the Atlantic thermocline. The thermocline also is sourced with water from box $d$ through diffusive upwelling ($q_U$). The strength of the downward branch of the AMOC is represented in the North Atlantic by $q_N$. This downwelling is dependent on the meridional density gradient between box $ts$ and box $n$, where the density is determined using a linear equation of state. Wind driven gyre transport is modelled by $r_N$ in the Northern Hemisphere, and $r_S$ in the Southern Hemisphere. Salinity is also affected by two surface freshwater fluxes, modelled as virtual salt fluxes. First, there is a symmetrical forcing $E_s$, i.e. this freshwater flux is the same for both hemispheres; and secondly, there is an asymmetrical forcing $E_a$ which results in interhemispheric differences. This last parameter can be viewed as a control parameter for the AMOC strength since it regulates the salinity of box $n$. The pycnocline depth is an important state variable in this model since several volume fluxes are dependent on it. This depth is dependent on four different volume fluxes going in and out of the two thermocline boxes $t$ and $ts$ ($q_e$, $q_{Ek}$, $q_U$, $q_N$).

The model provides a simple framework to study AMOC dynamics and has already been used to show both slow (Cimatoribus et al., 2014) and fast, noise-induced (Castellana et al., 2019; Jacques-Dumas et al., 2023) tipping of the AMOC.

## 2.2 Carbon cycle model

The carbon cycle model is derived from the equations of the SCP-M (O'Neill et al., 2019). The original SCP-M has two terrestrial carbon stocks, an atmosphere box, and 7 ocean boxes representing the global ocean. In the ocean multiple tracers are simulated that are important for the marine carbon cycle. In this study, we only simulate dissolved inorganic carbon (DIC), alkalinity (Alk) and phosphate ($PO_4$) in the ocean. All three tracers are affected by ocean circulation, have a riverine source and a sink to the sediments. DIC is affected by biological production and remineralization (soft tissue pump), the formation and dissolution of calcium carbonate ($CaCO_3$; carbonate pump), and gas exchange with the atmosphere. Alk is also affected by the carbonate pump, and $PO_4$ by the soft tissue pump. In this model, $PO_4$ is explicitly conserved, i.e. the source of $PO_4$ is equal to the sink of $PO_4$ at all times. DIC and Alk, however, can vary since the time dependent riverine influx is not necessarily equal to the sediment outflux.

The soft tissue pump is modelled using constant values of export production per box, and the remineralization in the water column follows a power law (Martin et al., 1987). The influence of the soft tissue pump on the cycling of $PO_4$ is modelled using a constant stoichiometric ratio. The formation of $CaCO_3$ is proportional to the export production times a constant rain ratio parameter. $CaCO_3$ is dissolved through the water column and in the sediments. This dissolution is dependent on the $CaCO_3$ saturation state, and a constant background dissolution. The gas exchange between the ocean and atmosphere is dependent on a constant piston velocity and the difference in $pCO_2$ between the two reservoirs. The riverine influx of $PO_4$ is constant, whereas the influx of DIC and Alk is dependent on atmospheric $pCO_2$.

## 2.3 Coupled model

The two models described in the previous section are coupled to form the model used in this study (Fig. 1). For this, several parameter assumptions had to be made, since the carbon cycle model requires more parameters than the AMOC model. First of all, the depth of boxes $n$ and $s$ is not given in Cimatoribus et al. (2014) but is necessary for the carbon cycle model. We assume these to be 300 m, and the total depth of the ocean is assumed to be 4000 m. Secondly, a first version of the model showed a too strong sensitivity of atmospheric $CO_2$ concentrations to AMOC tipping causing very low $CO_2$ concentrations on the AMOC off-branch. We therefore have included two additional boxes in the AMOC model representing the Indo-Pacific basin: box $ps$ for the surface ocean and box $pd$ for the deep ocean. In these boxes the same carbon cycle processes are present as in the Atlantic and Southern Ocean boxes of the model. Between these two boxes there is a bidirectional mixing term ($\gamma_1 = 30$ Sv), and the boxes are connected with the Southern Ocean through a Global Overturning Circulation (GOC; $\psi_1 = 18$ Sv), and gyre-driven exchange ($r_P = 90$ Sv). $\gamma_1$ and $\psi_1$ are taken from the SCP-M (O'Neill et al., 2019), and $r_P$ is based on the model of Wood et al. (2019). Both box $t$ and $ps$ receive DIC, Alk and $PO_4$ input through a riverine flux. The total riverine flux is modelled similarly as in the SCP-M and is partitioned over the two boxes based on the volume fraction of the Atlantic Ocean and the Indo-Pacific Ocean, meaning 20% of the riverine flux flows into box $t$, while the remainder flows into box $ps$.

The first coupling between the physical and the carbon cycle model is through the ocean circulation. The AMOC determined in the circulation model is used for the advective transport of the three tracers in the carbon cycle model. We have implemented additional couplings between the model and specific feedbacks within the carbon cycle model. Several of these feedbacks have been introduced into the SCP-M before (Boot et al., 2022).

Biological export production is constant in the SCP-M and therefore independent of available nutrients. This is a strong simplification of important processes in the real world that might not be valid in all cases. Therefore, we want to make the biological export production a function of nutrient availability. We do this by creating a dependency of the biological export production in the surface boxes to the amount of $PO_4$ advected into the specific surface box and therefore introducing a dependency on the ocean circulation

$$Z_i = (1 - \lambda_{BI}) \times Z_{i,base} + \lambda_{BI} \times (\sum_j (q_{j \to i} \times [PO_4^{3-}]_j) + P_{river}) \times \epsilon_i. \tag{1}$$

Here $Z_i$ represents the export production in surface box i, $\lambda_{BI}$ a parameter to switch between the default value of Z in box i ($Z_{i,base}$; $\lambda_{BI} = 0$) and the variable export production ($\lambda_{BI} = 1$). In addition, $q_{j \to i}$ represents the volume transport from box j into box i. $P_{river}$ the riverine influx of $PO_4$, which is only present in boxes $t$ and $ps$, and $\epsilon_i$ represents a biological efficiency term in box i. i represents all surface boxes, i.e. *n, t, ts, s* and *ps*. j can be any box and depends on the direction of the ocean circulation. In the text we will refer to this coupling as the BIO coupling. By using this coupling, a weaker (stronger) ocean circulation would result in a reduced (increased) influx of nutrients which causes a reduction (increase) in biological carbon export from the surface to the deep ocean. This affects carbon content in the surface ocean and carbon burial in the sediments.

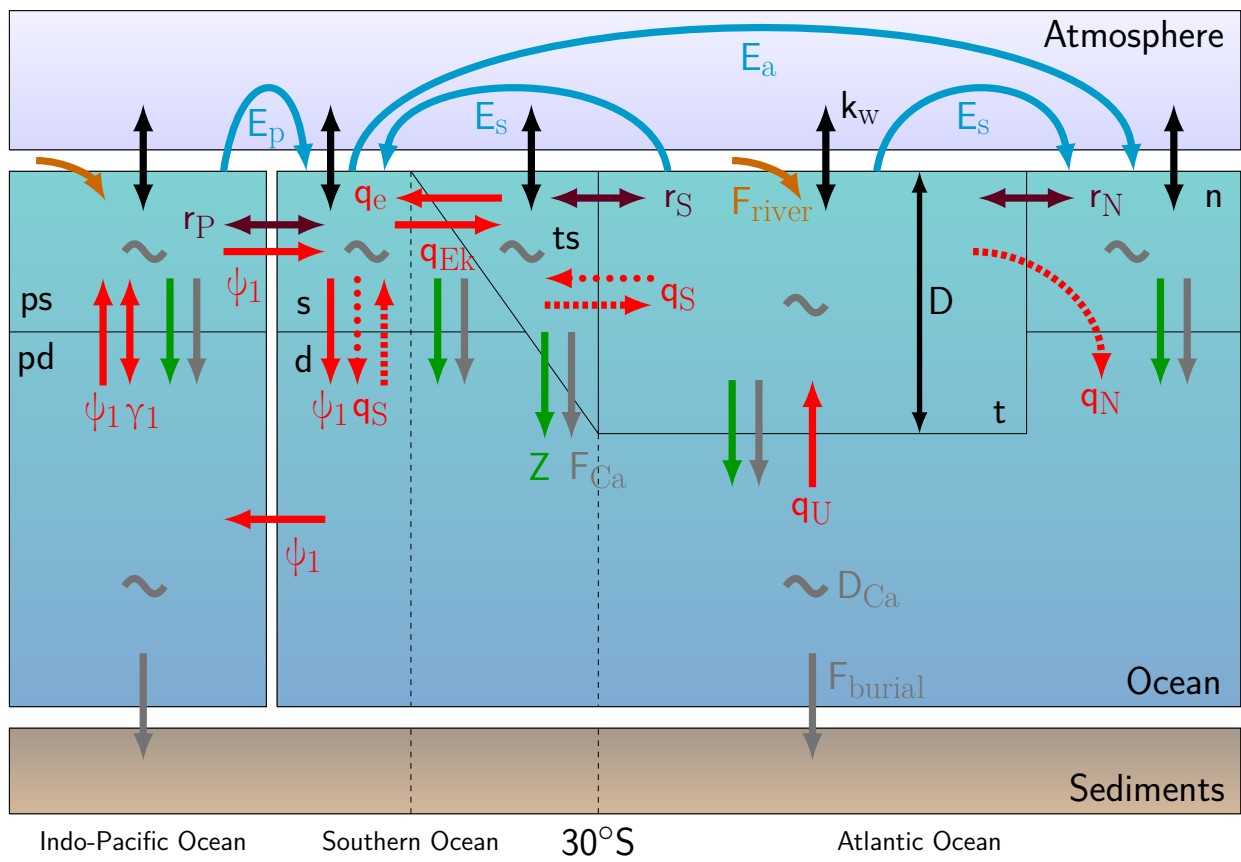

**Figure 1.** Box structure and processes simulated in the coupled circulation – carbon cycle model. Red arrows represent volume transports where dashed arrows are only present during an on-state, and dotted arrows only present during an off-state. The AMOC downwelling strength is represented by $q_N$ and is determined through $\eta \frac{\rho_n - \rho_{ts}}{\rho_0} D^2$, where $\eta$ is a hydraulic constant, $\rho$ represents the density in boxes n ($\rho_N$) and ts ($\rho ts$), and a reference density ($\rho_0$). D represents the thermocline depth. The purple arrows represent gyre exchange ($r_N$, $r_S$, and $r_P$), and blue arrows freshwater fluxes ($E_s$, $E_a$, and $E_p$). Carbon cycle processes that are represented are riverine input (orange), air-sea gas exchange (black; $k_w$), biological export production (green; Z), $CaCO_3$ rain (grey; $F_{Ca}$), $CaCO_3$ dissolution (grey; $D_{Ca}$), and sediment burial (grey; $F_{burial}$). Based on Castellana et al. (2019) and Boot et al. (2022).

We also introduce a coupling between the symmetric freshwater forcing $E_s$ and atmospheric $pCO_2$. This coupling is based on a fit to an ensemble of CMIP6 Earth System Models and is described in Section 3.1. We do this because we expect freshwater fluxes to change under different background climates (Galbraith and de Lavergne, 2019). The AMOC is dependent on $E_s$ and this coupling can therefore result into a changing AMOC under different $pCO_2$ values.

We allow the sea surface temperatures (SSTs) to vary with atmospheric $pCO_2$ following a logarithmic function and a climate sensitivity parameter, according to

$$T_i = T_{i,base} + \Delta T_i, \tag{2}$$

$$\Delta T_i = \lambda_T \times 0.54 \times 5.35 \ln(\frac{CO_2}{CO_{2,0}}). \tag{3}$$

Here i represents the different surface ocean boxes. By varying the parameter $\lambda_T$ we are able to change the climate sensitivity of the model. In this study we use a value of $\lambda_T = 0$ (default), $\lambda_T = 1$ ($CS_{LO}$) and a value of $\lambda_T = 2$ ($CS_{HI}$), representing SST warming of 0 K, 2 K and 4 K per $CO_2$ doubling. For the default values, sea surface temperature remains constant independent of atmospheric $pCO_2$ values. For surface air temperature in CMIP6 models, the response to a $CO_2$ doubling is between 1.8 and 5.6 K (Zelinka et al., 2020). When this coupling is used, the changes in SSTs will also change the density in the ocean circulation model. However, since we use a linear equation of state and the change of SST is homogeneous over all surface boxes, it does not influence the ocean circulation. In the text we will refer to this coupling as the $CS_{LO}$ and $CS_{HI}$ couplings for the low and high climate sensitivity cases, respectively. This coupling introduces a positive feedback: higher atmospheric $pCO_2$ values lead to warming of the SSTs, which reduces the solubility of $CO_2$ in the ocean, meaning that more $CO_2$ will remain in the atmosphere. This feedback might be important for states where the $CO_2$ concentration deviates strongly from $pCO_{2,0}$.

Lastly, we have introduced a coupling on the rain ratio (Eq. 4) making it dependent on the saturation state of $CaCO_3$ following

$$F_{Ca,i} = (1 - \lambda_F) \times F_{Ca,base} + \lambda_F \times 0.022(\frac{[Ca_i^{2+}][CO_3^{2-}]}{K_{sp,i}} - 1)^{0.81}, \tag{4}$$

where i represents the different surface ocean boxes. Similar to the biological coupling coefficient $\lambda_{BI}$, $\lambda_F$ is either 0 or 1, and including this feedback will introduce different rain ratios per box. This feedback is based on the work of Ridgwell et al. (2007) where the parameters 0.022 and 0.81 have been used as a calibration parameter in the GENIE-1 Earth System Model. In the text we will refer to this coupling as the FCA coupling. In this coupling, the rain ratio is increased if more carbonate is available, which represents higher calcification rates under such conditions. In our model this affects the transfer of carbon and alkalinity from the surface ocean to the deep ocean and the sediments. This feedback is included because it can be important on the long timescales we investigate here.

We have included additional couplings in the model that are described in Appendix A. They are not included in the main text since they do not show large effects on the results. In the main text only the couplings described above are used. We refer to the couplings as BIO for the biological coupling (BIO), $E_s$ for the $E_s$-coupling described in Section 3.1, FCA for the rain ratio coupling, $CS_{LO}$ for a low climate sensitivity and $CS_{HI}$ for a high climate sensitivity.

As explained in the sections above, we have altered the box structure of both models, and included several couplings and feedbacks in the model. These changes in the model can change the model dynamics compared to the original models, i.e.

the AMOC box model (Cimatoribus et al., 2014; Castellana et al., 2019) and the SCP-M (O'Neill et al., 2019). Compared to the literature (Cimatoribus et al., 2014; Castellana et al., 2019), AMOC dynamics in our seven box model are very similar to the dynamics in the original five box model. The new box structure and ocean circulation change the carbon cycle quite a bit compared to the original SCP-M. To account for this, we have retuned the model before use such that atmospheric $pCO_2$ is around pre-industrial values as detailed in Section 2.4. However, the most important aspects of the SCP-M are the carbon cycle dynamics. When no couplings are used, these are still the same. When couplings are introduced, the model is changed further and the effects of these changes are one of the aspects we investigate in this study.

## 2.4 Solution method

The coupled model is a system of 30 ODEs (four tracers per box, the pycnocline depth and atmospheric $pCO_2$) of the form

$$\frac{d\mathbf{u}}{dt} = f(\mathbf{u}(t), \mathbf{p}). \tag{5}$$

Here $\mathbf{u}$ is the state vector (containing all the dependent quantities in all boxes), f contains the right-hand-side of the equations and $\mathbf{p}$ is the parameter vector. To solve this system of equations we use the continuation software AUTO-07p (Doedel et al., 2007). Both the AMOC model (Cimatoribus et al., 2014), and the SCP-M (Boot et al., 2022) have already been implemented in this software. AUTO enables us to efficiently compute branches of stable and unstable steady state solutions under a varying control parameter. Furthermore, it allows for detection of special points such as saddle-node bifurcations, here important for determining the multiple equilibria window of the AMOC.

One of the requirements of AUTO is that the Jacobian of the system (5) is non-singular at non-bifurcation points. To achieve this, we use explicit conservation equations to eliminate the ODEs of the deep Atlantic box (d). Both the conservation equation of salt and $PO_4$ are already explicitly included into the model. However, as described previously, this is not the case for DIC and Alk. Therefore, we have to introduce extra ODEs describing the change in total carbon and alkalinity in the system. The change in total carbon (DIC + atmospheric $CO_2$) and Alk in the atmosphere-ocean system can be captured as the sum of riverine influx and the sediment outflux. The riverine influx is a function of atmospheric $pCO_2$ and represents the weathering of silicate and carbonate rocks i.e.,

$$C_{river} = W_{carb,c} + (W_{carb,v} + W_{si}) \times CO_2^{atm}. \tag{6}$$

The sediment outflux of DIC is determined by the sum of the soft tissue and the carbonate pumps over the entire ocean. In this model, all produced organic matter is also remineralized in the water column, causing the contribution of the soft tissue pump to be negligible resulting in

$$C_{sed} = C_{river} \times V_t + \sum_{i=1}^{7} (C_{carb,i} \times V_i). \tag{7}$$

Since the change in alkalinity in the system is proportional to the change in total carbon, only one extra ODE is necessary. By eliminating the ODEs for the deep box and introducing the ODE for total carbon in the ocean-atmosphere system, AUTO eventually solves a system with 27 ODEs.

The use of AUTO made it necessary to make changes in the carbonate chemistry of the carbon cycle model. In the original SCP-M a simple time dependent function is used where the pH of timestep k-1 is used as an initial guess for timestep k (Follows et al., 2006). As long as the changes per time step remain relatively small, this scheme is sufficiently accurate. However, due to our solution method, in which steady states are calculated versus parameters, this function is not suitable for this study. Therefore, we have chosen a simple 'text-book' carbonate chemistry (Williams and Follows, 2011; Munhoven, 2013) where Alk is assumed to be equal to carbonate alkalinity ($Alk_{carb}$ = [$HCO_3^-$]+[$CO_3^{2-}$]). This method is less accurate and leads to higher pH values (Munhoven, 2013) and lower atmospheric $pCO_2$ values (Boot et al., 2022). To address the lower resulting atmospheric $pCO_2$ values we have increased the value of the constant rain ratio from 0.07 as used in the original SCP-M to 0.15.

AUTO has three parameters that determine the accuracy of the solution. The absolute and relative accuracy are set to a base value of $10^{-6}$, but sometimes a higher accuracy is used. The accuracy for the detection of special points (e.g. saddle-nodes and Hopf bifurcations) is set to $10^{-7}$.

## 3   Results

### 3.1   CMIP6 freshwater fluxes

The freshwater fluxes $E_s$ and $E_p$ used in the model are constrained using results from a CMIP6 ensemble. For this we use 28 different CMIP6 models forced with a 1% increase per year in atmospheric $CO_2$ concentrations ('1pctco2'). We integrate the variables 'wfo' (water flux) and 'vsf' (virtual salt flux) over the regions representing the Atlantic thermocline (Atlantic basin between 30°S and 50°N) and the Indo-Pacific basin (the rest of the ocean north of 30°S and south of 66°N) in the coupled box model. Based on these 28 models we determine a multimodel mean and we are able to constrain both $E_p$ and $E_s$. For a full list of the models and used ensemble members see Table A1.

Fig. 2a shows that most models, and the multimodel mean, show no, or at most a very weak relation between $E_p$ and atmospheric $pCO_2$, whereas there seems to be a relation between $E_s$ and atmospheric $pCO_2$. For $E_p$ we will use the mean value over the entire simulation (0.99 Sv). For $E_s$ we will use as a default value 0.39 Sv since this is the value of $E_s$ at $pCO_{2,0}$ (320 ppm). Furthermore, we introduce an additional coupling in the model where we implement $E_s$ as a function of atmospheric $pCO_2$ based on a logarithmic fit to represent the relation between $E_s$ and atmospheric $pCO_2$ present in the CMIP6 ensemble. This relation is modelled as:

$$E_s = (1 - \lambda_E) \times E_{s,base} + \lambda_E \times (-0.142 + 0.092 \times \ln(CO_2)) \tag{8}$$

Here $\lambda_E$ is a parameter controlling whether the coupling is used ($\lambda_E = 1$) or the default value of $E_{s,base}$ (0.39 Sv) is used ($\lambda_E = 0$). Compared to earlier versions of the model we will use a different default value for $E_s$. In previous studies values

of 0.25 Sv (Cimatoribus et al., 2014) and 0.17 Sv (Castellana et al., 2019) have been used. Here we choose the default value based on the value of $E_s$ at an atmospheric $pCO_2$ value of 320 ppm ($pCO_{2,0}$) in the CMIP6 fit. The value of 0.39 Sv is of the same order as seen in the HOPAS4.0 dataset based on satellite observations (Andersson et al., 2017). This dataset shows a net freshwater flux of 1 Sv averaged over the period 1987-2015 into the region representing the thermocline box, which results in an $E_s$ value of 0.5 Sv. In the text we will refer to this coupling as the $E_s$ coupling. Note that this fit does not necessarily

represent a direct causal relation between atmospheric $pCO_2$ and the freshwater flux. Surface temperature could also play an important role here. However, we have included effects of temperature changes in relation to $CO_2$ through the CS coupling. The $E_s$ coupling is responsible for the changes in salinity related to different $CO_2$ concentrations.

We have made two important choices for using these CMIP6 constrained freshwater fluxes. First of all, we set the freshwater

transport through the atmosphere from the Atlantic to the Indo-Pacific basin to 0. There are studies showing there is moisture transport between the two basins through the atmosphere (e.g., Dey and Döös, 2020), but it is challenging to constrain this flux from Earth System Models. However, in our model set up, the exact value of this flux is not relevant for our results. The total freshwater flux integrated over the Indo-Pacific basin diagnosed from the CMIP6 ensemble is independent from the moisture transport between the Atlantic and Indo-Pacific basin. By rescaling the freshwater flux from the Indo-Pacific basin (box *ps*)

to the Southern Ocean (box *s*) we can set the freshwater flux from the Atlantic to the Indo-Pacific to 0 without changing the AMOC dynamics. Tests where this flux was not set to 0, but net evaporation out of boxes *t* and *ps* were kept constant show this. The only effect of this freshwater transport is a shift of the diagram along the $E_a$ axis and a small effect on atmospheric $pCO_2$ of a couple of ppm due to salinity changes.

The second choice we have made is that the net evaporation from the Atlantic thermocline is symmetrically divided over the

northern and southern high latitudes. For this model, the exact direction of the freshwater flux out of box *t* is irrelevant. What is relevant is the total freshwater flux at each surface box. Through this we can see that the asymmetric freshwater flux, $E_a$, creates an asymmetry in freshwater forcing over the Atlantic basin. Through this, $E_a$ creates the asymmetry that is potentially more realistic. Since we use $E_a$ as our control parameter in the continuations, we do not need to constrain this parameter.

## 3.2 The AMOC multiple equilibria window

We use several different model configurations that are differentiated on feedbacks and couplings included (see Table 1). We use these different configurations to show the effect on non-linear feedbacks on the MEW. Note that different couplings (see Appendix A) and different combinations of couplings are possible, but we have chosen to use incremental steps in including new couplings to keep the results as simple as possible. We also chose to limit the number of couplings in the main text for the same reason, i.e. to keep it as simple as possible, and because these couplings have the strongest effect on the model results.


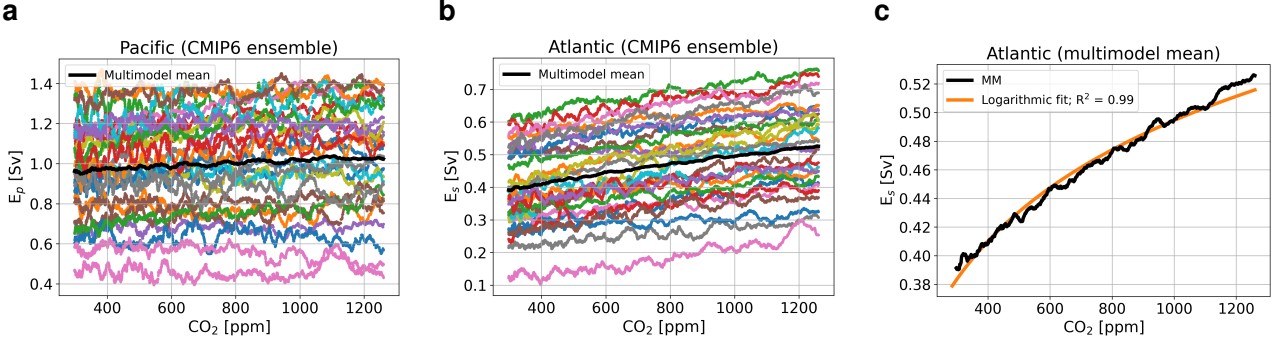

**Figure 2.** (a) Net evaporation from the Indo-Pacific basin representing the freshwater flux $E_p$ in Sv for the CMIP6 ensemble with the multimodel mean in black. (b) As in (a) but for the freshwater flux $E_s$. (c) The multimodel mean for $E_s$ in black with a logarithmic fit in orange.

**Table 1.** Overview of the used cases. The left column represents the name of the case. The other columns represent whether a coupling denoted in the top row is used in the case mentioned in the first column by indicating the $\lambda$ parameter associated to the coupling. For $\lambda_T$ the value represents the strength of the coupling. The quantity $\lambda_{BI}$ refers to Eq. 1 (biological coupling), $\lambda_E$ to Eq. 8 ($E_s$ coupling), $\lambda_F$ to Eq. 4 (rain ratio feedback), and $\lambda_T$ to Eq. 3 (temperature).

| Case name | $\lambda_{BI}$ (Eq. 1) | $\lambda_E$ (Eq. 8) | $\lambda_F$ (Eq. 4) | $\lambda_T$ (Eq. 3) |
|---|---|---|---|---|
| REF | 0 | 0 | 0 | 0 |
| BIO | 1 | 0 | 0 | 0 |
| $E_s$ + BIO | 1 | 1 | 0 | 0 |
| $E_s$ + BIO + FCA | 1 | 1 | 1 | 0 |
| $E_s$ + BIO + FCA + $CS_{LO}$ | 1 | 1 | 1 | 1 |
| $E_s$ + BIO + FCA + $CS_{HI}$ | 1 | 1 | 1 | 2 |

In our simulations we define the MEW as the range between the two saddle-node bifurcations which can include both stable and unstable branches. In Fig. 3 typical bifurcation diagrams for the AMOC strength (Fig. 3a) and atmospheric pCO$_2$ (Fig. 3b) versus $E_a$ are shown. Fig. 3 specifically shows the configuration where the biological coupling, i.e. where biological export production is dependent on ocean circulation, is used (case BIO). Bifurcation diagrams of the other model configurations
discussed here can be seen in Fig. A1 and are very similar to the diagrams shown in Fig. 3.

The bifurcation diagrams show that to be able to simulate both the on- and off-branch, it is vital that the BIO coupling is used. When this coupling is not used, PO$_4$ concentrations will become negative in the surface ocean under a collapsed AMOC regime. This behavior is illustrated in Fig. A1a, b for case REF. In case REF the off-branch (with negative PO$_4$) is not shown (Fig. A1a, b), while for case BIO the full bifurcation diagram with two saddle-node bifurcations is plotted (Fig. 3). The reason
that PO$_4$ concentrations become negative is that as the AMOC strength declines, less PO$_4$ is advected into box n decreasing

$PO_4$ concentrations there. As in case REF biological export production is constant, at some point the sink (i.e. mainly biological export production) becomes larger than the source (i.e. advection of $PO_4$) and $PO_4$ concentrations will become negative. This shows that the model without the BIO coupling is unable to capture the carbon cycle of a collapsed AMOC state because of missing processes, most notably the reduction in biological export production under increased nutrient limitation. In Fig. 3b

we can also see the effect of AMOC tipping on atmospheric $pCO_2$. On both the on- and the off-branch, atmospheric $pCO_2$ values are relatively constant and the difference between the branches is approximately 25 to 40 ppm depending on the case considered, values that are of the same order as values reported in more complex models (Gottschalk et al., 2019). It is good to note here that we do not expect the same response as those found in more complex models, since we employ a steady state approach while more complex models use transient simulations that are not yet in equilibrium. However, we would not expect

a much larger response in magnitude and since our response is of similar order as that in Gottschalk et al. (2019), we have confidence that the model is suitable for our application.

To explain the lower $pCO_2$ values on the off-branch we consider the constraint in the model on total carbon content in the ocean-atmosphere system. In steady state, total carbon content in the ocean-atmosphere system, is not allowed to change. Note that this does not mean that for every $E_a$ value total carbon content is the same. Different $E_a$ values correspond to a slightly

different total carbon content in the ocean-atmosphere system, but for each $E_a$ value $\frac{dTC}{dt}$ is equal to 0. Terrestrial and soil carbon are not considered in this model. This means that the riverine input and sediment outflux of DIC must balance for each value of $E_a$ to keep the total carbon content constant. In our model, the sediment outflux is a function of the saturation state of $CaCO_3$ and $CaCO_3$ flux which is a function of the rain ratio (constant in non-FCA cases) and the export production. However, in the AMOC off state, the saturation state of $CaCO_3$ in the ocean is in every box larger than 1, meaning that there

is no saturation driven dissolution of $CaCO_3$ and the sediment outflux is purely a function of the export production and a constant background dissolution rate. In an AMOC off-state, nutrient advection is relatively low causing a reduction in export production, and therefore a smaller sediment outflux. In steady state, the riverine influx must balance this small outflux, which is only possible by decreasing atmospheric $pCO_2$ values.

From the 6 cases considered here (Table 1) we can see the effect of the individual couplings. As described earlier, the

biological coupling is necessary to determine the off-branch but does not influence the bifurcation diagrams otherwise. Adding the $E_s$ coupling ($E_s$ as function of atmospheric $pCO_2$) alone does not affect the dynamics of the model (Fig. A1c, d) too much since $CO_2$ concentrations are close to $CO_{2,0}$. The rain ratio coupling (FCA; variable rain ratio dependent on $CaCO_3$ saturation state) decreases atmospheric $CO_2$ concentrations by 35 ppm and slightly increases the difference in $CO_2$ concentration between the on- and off-branch (Fig. A1f). This coupling decreases the atmospheric $CO_2$ concentrations because under these settings,

the FCA coupling leads to a lower rain ratio compared to a constant rain ratio. As a result, burial of carbon in the sediments is reduced, meaning that also the river influx is reduced, which can only be caused by a lower atmospheric $CO_2$ concentration. The climate sensitivity coupling increases this effect by changing the solubility of $CO_2$ in the surface ocean, with a larger effect for the higher climate sensitivity (Fig. A1h, j). In the cases using the rain ratio, the potential of the $E_s$-coupling becomes visible. In these cases, atmospheric $pCO_2$ values deviate more from $pCO_{2,0}$ and therefore have a larger effect on $E_s$. When $E_s$

differs from the default value (0.39 Sv), both saddle-node bifurcations move to different $E_a$ values.

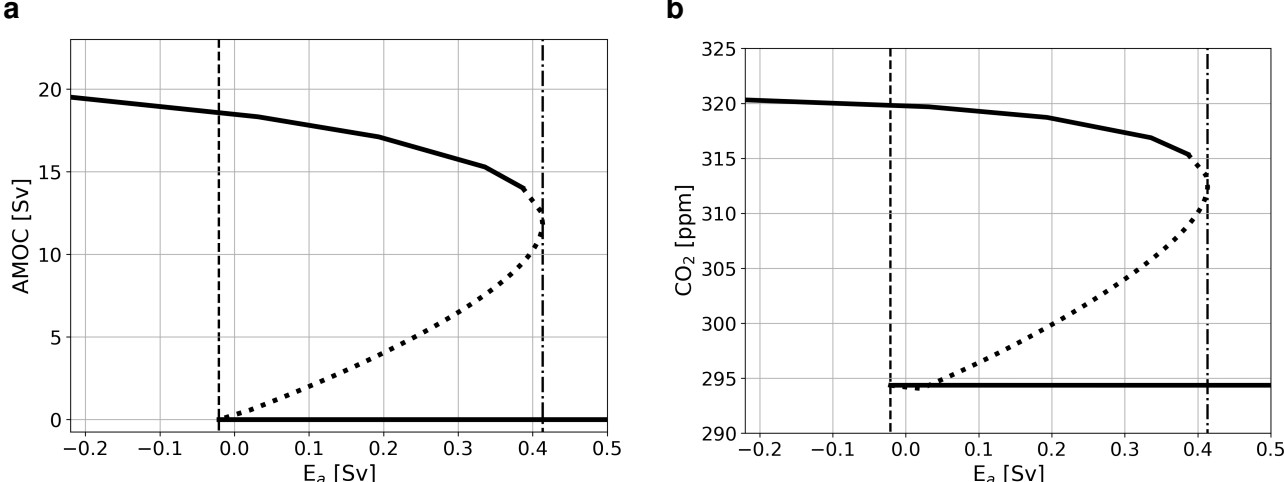

**Figure 3.** Bifurcation diagram showing the sensitivity of the AMOC and atmospheric $pCO_2$ to $E_a$. Solid lines represent stable steady state solutions, dotted lines represent unstable solutions, vertical dash-dotted lines represent the location of the saddle-node bifurcation on the on-branch, and vertical dashed lines the location of the saddle-node bifurcation on the off-branch. The case presented here is the one where the biological coupling is used, i.e. case BIO. Bifurcation diagrams of other cases discussed in the main text can be found in Fig. A1.

To explain the movement of the saddle-node bifurcations, we consider the sensitivity of the model to $E_s$ (Fig. 4). In Fig. 4 the location of the saddle-node bifurcations on both the on- and the off-branch are shown versus the value of $E_s$. This figure shows that as $E_s$ increases, the MEW also increases. The default value used for cases REF and BIO for $E_s$ is 0.39 Sv. The CMIP6 $CO_2$-dependent fit (8) results in a slightly smaller value. Due to decreased $E_s$, the thermocline becomes fresher, and

in combination with the salt-advection feedback, this leads to a smaller meridional density gradient and therefore a weaker AMOC. Furthermore, decreased $E_s$ decreases the net evaporation over the Atlantic, given by ($E_s$-$E_a$) and this means that a smaller $E_a$ is necessary to tip the AMOC. On the off-branch, a smaller $E_s$ results in salinification of the *ts* box and a less negative freshwater flux ($E_a$) is needed to decrease the meridional density gradient and reinvigorate the AMOC. For cases with the FCA feedback, it reduces the MEW by moving the off-branch saddle-node bifurcation to larger values of $E_a$, and the

saddle-node bifurcation on the on-branch to smaller values, which can be explained by the fact that $CO_2$ is smaller than $CO_{2,0}$ and therefore $E_s$ is smaller than $E_{s,base}$ in (8).

In the bifurcation diagrams in Fig. 3 and Fig. A1 we find that the solution on the on-branch becomes unstable before passing the saddle-node bifurcation. This change in stability can be explained by the presence of a subcritical Hopf bifurcation in the circulation model. The internal oscillation corresponding to this Hopf bifurcation is unstable and has a multidecadal periodicity.

In this study we are only interested in the MEW of the AMOC, and we therefore do not consider the Hopf bifurcation further.

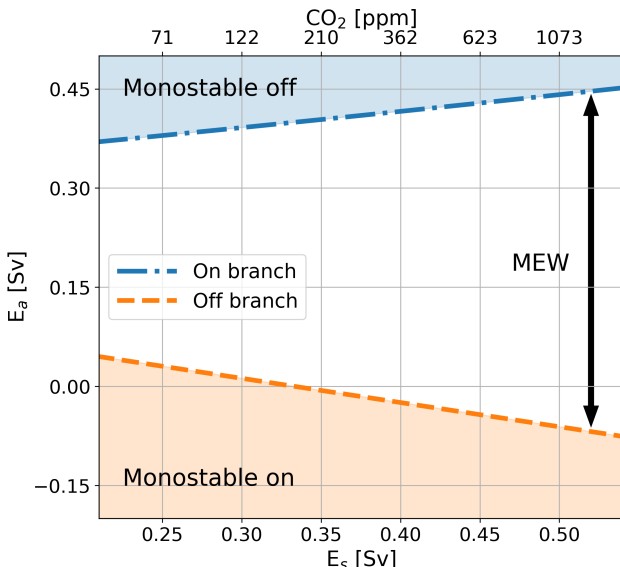

**Figure 4.** $E_a$-value corresponding to the saddle-node bifurcation on the on-branch (dash-dotted blue line) and the off-branch (dashed orange line) for different values of $E_s$ in Sv (bottom x-axis). The area above the blue-dotted line represents the monostable off state, the area below the orange line the monostable on state, and the area in between the MEW. The top x-axis represents the $CO_2$ values corresponding to the $E_s$-values following the fit (Eq. 8); note that this axis is non-linear. The results are based on the dynamical ocean model only where the value for $E_s$ has been changed.

### 3.3 Sensitivity to total carbon content

Over the Cenozoic, both the AMOC (Lynch-Stieglitz, 2017) and total carbon content in the ocean-atmosphere system have varied (Zeebe et al., 2009; Caves et al., 2016). In Caves et al. (2016) it is suggested that total carbon content has varied between 24,000 PgC and 96,000 PgC. In the previous section, the model was studied with approximately 40,000 PgC in the global system. In this section, we analyze how the sensitivity of the AMOC MEW changes under different total carbon contents in the model. To test the sensitivity, we remove approximately 4,000 (-10%) PgC, and add approximately 4,000 (+10%), 10,000 (+25%) and 20,000 (+50%) PgC. We do this for the cases considered in Section 3.2 excluding case REF (Fig. 5).

In case BIO there is no change in the MEW, which is to be expected since there is no back coupling from the carbon cycle model to the AMOC model, and the AMOC solution is therefore independent of the carbon cycle. We see only the effect of total carbon content on atmospheric $pCO_2$ values. When carbon is removed, the $CO_2$ concentrations at the saddle-node bifurcation both decrease. However, when carbon is added, only the saddle-node bifurcation on the on-branch has higher $CO_2$ concentrations, independent of whether 4,000, 10,000 or 20,000 PgC is added. We see a similar pattern for the $E_s$ + BIO case, but here the MEW increases for larger total carbon content due to the different $CO_2$ concentrations at the saddle-node bifurcations. The cases including the rain ratio feedback show a different pattern. Here, the $CO_2$ concentrations at both saddle-

node bifurcations are dependent on the amount of carbon added to the ocean-atmosphere system, i.e. the higher the content, the higher the $CO_2$ concentrations at the saddle-node bifurcations (Fig. 5b). This influences the value of $E_s$ at the saddle-node bifurcations (Fig. 5c), which increases the MEW for increasing carbon content (Fig. 5a). The MEW shift increases when the climate sensitivity coupling is used ($CS_{Lo}$ and $CS_{Hi}$), with a larger response for the higher sensitivity ($CS_{HI}$). Another effect visible in the cases using the FCA feedback is the difference in $CO_2$ concentration between the on- and the off-branch increases

as total carbon content increases. This effect is larger when climate sensitivity is increased.

We can explain the behavior of the MEW in the $E_s$ + BIO case by looking at the atmospheric $pCO_2$ values, and therefore also $E_s$, at the saddle-node bifurcations, which are similar for the three high total carbon cases. However, when the rain ratio feedback is used, we see that the MEW keeps increasing for larger carbon contents since also the atmospheric $pCO_2$ increases. We can explain the difference between $E_s$+BIO and the cases where the rain ratio feedback is used by the constraint on total

carbon in the ocean-atmosphere system. In $E_s$+BIO, biological export production in the Atlantic is mainly a function of the AMOC strength, whereas in the $E_s$+BIO+FCA case it is also dependent on the $CaCO_3$ saturation state which is coupled to atmospheric $pCO_2$ through the pH of the surface ocean. This leads to a larger outflux of DIC and Alk to the sediments, which, in steady state, needs to be balanced by a higher influx of DIC and Alk through the riverine flux, which can only be achieved by increasing atmospheric $pCO_2$.

A second result for the cases with the rain ratio feedback is that the $CO_2$ concentration difference between the on- and off-branch increases for higher total carbon content. As we increase total carbon content in the system, the rain ratio increases on both the on- and the off-branch because the saturation state of $CaCO_3$ increases. Due to non-linearities in the carbonate chemistry, the more carbon is present in the system, the larger the difference in rain ratio between the two branches. This explains why the difference between the on- and off-branch increases as total carbon content increases in the system.

**4  Summary and discussion**

In this paper we investigated the multiple equilibria window (MEW) of the AMOC in a coupled ocean circulation-carbon cycle box model. When freshwater forcing is coupled to atmospheric $pCO_2$ using a CMIP6 multi-model fit equation (8) above, the MEW changes slightly due to a dependency on atmospheric $pCO_2$. We also assessed the sensitivity to total carbon content in the system and found that the MEW is larger with more carbon in the system due to a shift of both the on- and off-branch

saddle-node bifurcations. These results show the potential of the marine carbon cycle to influence the MEW of the AMOC.

Two processes explain the results on the MEW: (1) the balance between the riverine flux and sediment flux that constrains atmospheric $pCO_2$ (first two panels in Fig. 6a, b); and (2) the sensitivity of the AMOC to $E_s$ (last panel in Fig. 6a, b). These clear and plausible mechanisms are more important than the precise quantitative estimates and are summarized in Fig. 6. In the model, atmospheric $pCO_2$ is dependent on the ocean circulation through the effect of export production on the burial of DIC and

Alk in the sediments. In steady state, this burial needs to balance the riverine influx which is dependent on atmospheric $pCO_2$. When the $E_s$-coupling is used, $E_s$ is dependent on atmospheric $pCO_2$, and the ocean circulation is dependent on $E_s$, creating a feedback loop (Fig. 6). If the $CO_2$ concentration in the atmosphere is larger than $CO_{2,0}$, the MEW increases, while it decreases

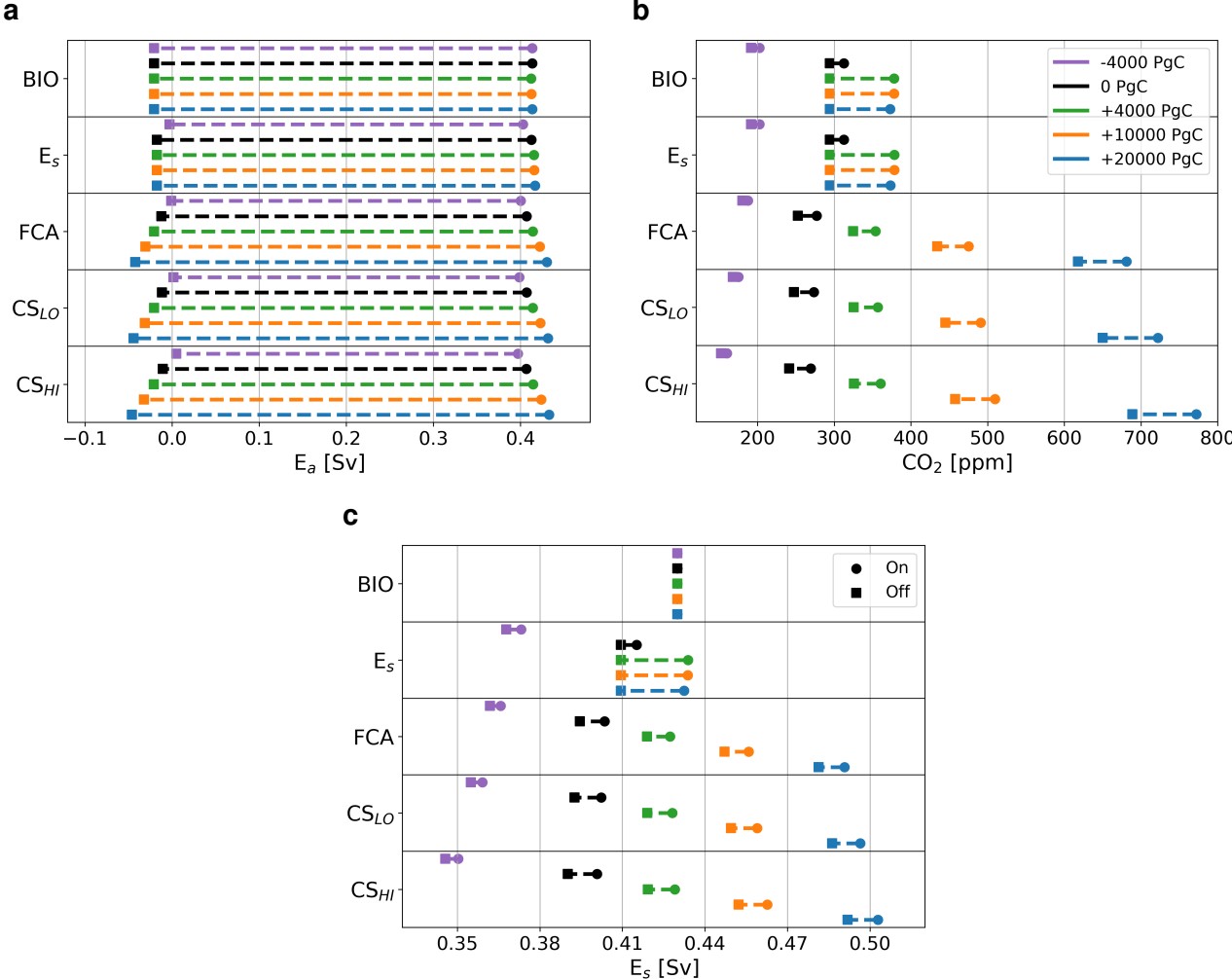

**Figure 5.** Panel a shows the location of the saddle-node bifurcations versus $E_a$ in Sv, panel b shows the corresponding $CO_2$ concentration in ppm, and c shows the corresponding value of $E_s$ in Sv. The top row of the figure represents case BIO, the second row case $E_s$ + BIO, and the middle row case $E_s$ + BIO + FCA, the fourth row case $E_s$ + BIO + FCA + $CS_{LO}$, and the bottom row $E_s$ + BIO + FCA + $CS_{HI}$. Square markers represent the location of the saddle-node bifurcation on the off-branch and round markers the location of the saddle-node bifurcation on the on-branch for cases where 4000 PgC is removed (purple), the default carbon content (black), 4000 PgC is added (green), 10,000 PgC is added (orange) and where 20,000 PgC is added (blue). Note that these values lie well in between 24,000 PgC and 96,000 PgC, the range of total carbon content throughout the Cenozoic as suggested by Caves et al. (2016), and the default total carbon content is approximately 40,000 PgC.

if it is smaller than $CO_{2,0}$. This results in that when atmospheric $pCO_2$ is high, so is $E_s$ which results in a stronger AMOC on the on-branch. As a consequence, export production is increased and there will be a larger outflux of carbon and alkalinity

through the sediments, which is balanced by a high influx of carbon through the rivers, consistent with high atmospheric $pCO_2$ values. Of the feedbacks that we have implemented, only the rain ratio feedback (FCA) affects this mechanism because it directly influences the sediment outflux and makes the carbon cycle less sensitive to the ocean circulation. Also the $E_s$ - $pCO_2$ fit used in this study is important. We acknowledge that it is difficult to assess the validity of the CMIP6 $E_s$-$pCO_2$ fit since that fit is based on a transient simulation with a strong forcing. However, longer (i.e. more than 3000 year) simulations by Galbraith and de Lavergne (2019) show a similar, actually slightly stronger relation than the one used in this study.

Vital in this mechanism is the riverine flux that is a linear function of atmospheric $pCO_2$. The linear function we have used in this study is based on the SCP-M (O'Neill et al., 2019), which is based on earlier work by Toggweiler and Russell (2008). In LOSCAR (Zeebe, 2012), a model of similar complexity, the riverine flux is based on a power law. However, this function is defined such that atmospheric $pCO_2$ converges to a preset value over time which makes it unsuitable for our study. There are models with more complex weathering terms including effects of temperature and vegetation, e.g. COPSE (Bergman et al., 2004) and GEOCARB-SULF (Royer, 2014), but these are too complex for our model. We could replace the linear parameterization also with non-linear ones. Powers larger than one will decrease the sensitivity of the model to changes in the burial of $CaCO_3$ in the ocean, and powers smaller than one will increase the sensitivity of the model. Given that the model does not seem to be very sensitive to non-linear feedbacks in the carbon cycle, we do not expect very different behavior if a non-linear parameterization is used.

The results here can be relevant when studying climate transitions in past and future climates as mechanisms how AMOC stability can depend on background climate and atmospheric $pCO_2$ values are identified. Previous work focused on the Pleistocene suggest an influence of atmospheric $pCO_2$ on the stability structure of the AMOC through temperature (Sun et al., 2022) and moisture transport (Zhang et al., 2017). In our model, there is no direct effect of temperature changes on the AMOC strength, but the $E_s$-coupling used here is similar to the moisture transport described in Zhang et al. (2017). The only difference is that this moisture transport is directly to the Pacific basin in their study, whereas in our model we rescale freshwater fluxes to set this direct flux to 0.

We have used a model that provides a simple framework for studying AMOC dynamics that allows us to efficiently test the concept of AMOC stability in a wide range of parameter values. However, a limitation is that in the model temperature is not a state variable, based on the assumption that the timescales of salinity variations is longer than that of temperature and thus dominant in steady state. This means that the AMOC strength in our model is not influenced by changes in temperature, which is a caveat of this study. Under high carbon content in the ocean-atmosphere system, this might not be valid. However, we have explored relatively small changes in the total carbon content and the mechanisms presented here are also valid for this smaller range, suggesting that the main mechanism presented in this study is at least valid for small changes in the total carbon content. A recent study (van Westen et al., 2024) where the original box model of Castellana et al. (2019) is extended with dynamical temperature equations shows that under present day conditions the MEW hardly changes after this extension of the model. Willeit and Ganopolski (2024) show that under higher $CO_2$ concentrations, the MEW increases in the EMIC CLIMBER-X. Note that this is done without interactive carbon cycle, so this is just the response of the AMOC to warmer climates in a more

**a**

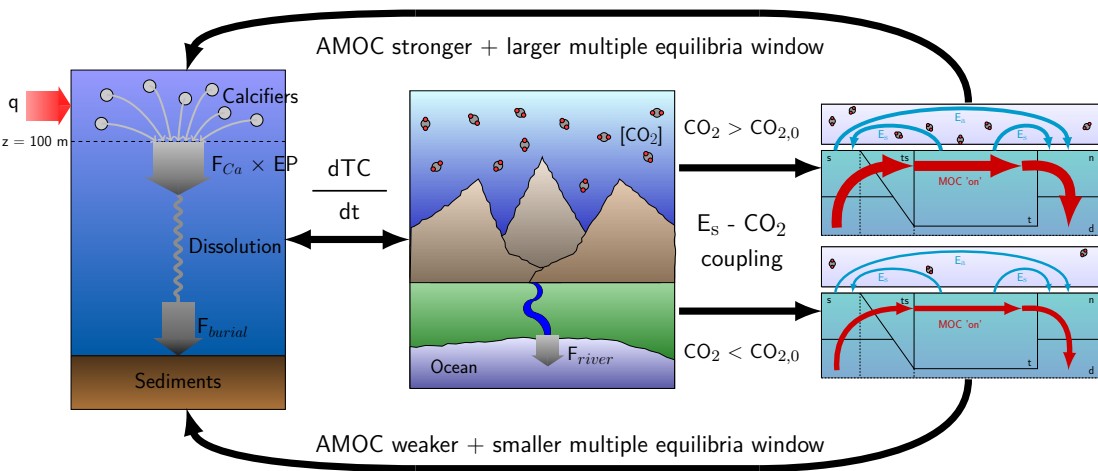

**b**

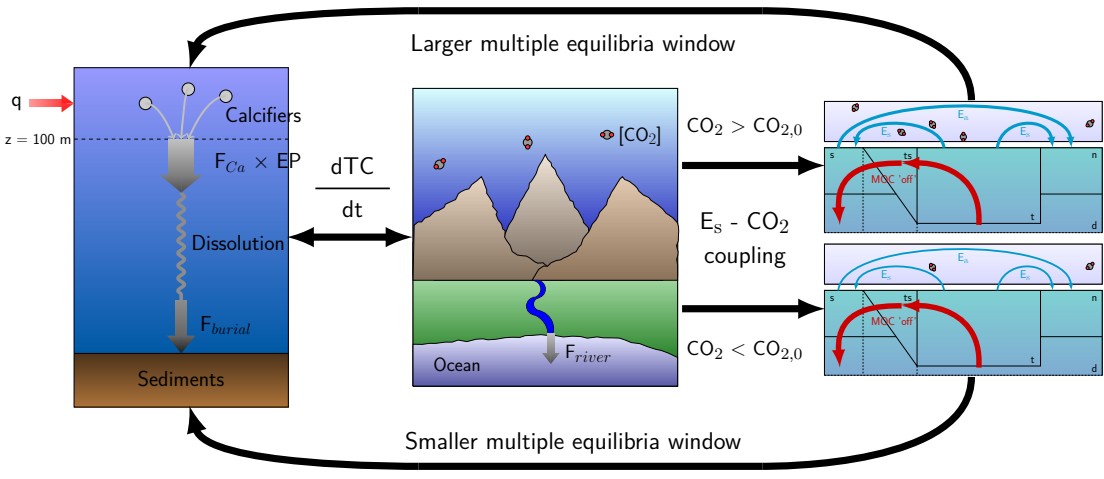

**Figure 6.** Illustrations of the main mechanisms affecting atmospheric $pCO_2$ and AMOC stability. Panel a shows the mechanisms for the on-branch. A strong AMOC increases export production through increased nutrient advection (left panel), which is accompanied by a high atmospheric $pCO_2$ due to the necessary balance between the river influx and sediment burial (middle panel). If the $CO_2$ concentration is larger (smaller) than $CO_{2,0}$ than the AMOC will strengthen (weaken) and the MEW increases (decreases) (right panels). Panel b shows the mechanisms for the off-branch. The absence of an AMOC decreases export production through decreased nutrient advection (left panel), accompanied by a low atmospheric $pCO_2$ (middle panel). When $pCO_2$ is larger (smaller) than $pCO_{2,0}$ the MEW increases (decreases) (right panel). TC represents total carbon in the ocean-atmosphere system, EP the export production, and $F_{Ca}$ the rain ratio.

complex model than the one used in this study. Based on these two studies, we do not expect that the MEW shift described in this study is fully compensated for when temperature is a state variable.

In van Westen et al. (2024) the original box model was also extended with a parameterization representing the effects of sea ice on the AMOC. This parameterization is based on hysteresis experiments using the Community Earth System Model (van Westen and Dijkstra, 2023). Sea-ice insulation effects create a new state in the model with a weak AMOC that extends from the off branch towards lower values of $E_a$. This effectively increases the MEW in the model, showing that sea ice can play an important role in the ocean dynamics of the model. The weak state is expected to disappear in warmer climates because of melting of the sea ice. However, the changes in ocean dynamics do not necessarily impact the mechanisms summarized in Fig. 6, and we therefore believe that including sea-ice effects would not change the conclusions of our study.

Though not a limitation in the model, it is good to note that the range of timescales in the carbon cycle model is larger than in the circulation model, which does not affect our results but does affect the time dependent response of the system. As time-dependent effects are not considered, it is difficult to compare our results to existing studies in literature since these commonly use time integration. Studies using Earth System Models on multidecadal to centennial timescales expect that under climate change atmospheric $pCO_2$ values increase following reduced mixing in the North Atlantic (Boot et al., 2023b), or a weakening of the AMOC (Boot et al., 2024; Zhang et al., 2024). On millennial timescales, most studies show an increase in atmospheric $pCO_2$ after an AMOC weakening (Zickfeld et al., 2008; Gottschalk et al., 2019), but the mechanisms are dependent on the model and the set up of the simulations. The sign of response in atmospheric $pCO_2$ in most studies on the multidecadal to millennial timescales is at odds with what we found. However, this can be explained that the final response in our study is mostly dominated by longer timescale processes, i.e. the balance between weathering and burial of carbon in sediments. Our finding that the MEW increases under higher $CO_2$ concentrations is supported by results from CLIMBER-X (Willeit and Ganopolski, 2024). However, as noted earlier, this study does not use an interactive carbon cycle and the increase in MEW is caused only by the response of the AMOC to a warmer climate.

Our work also holds implications for assessing AMOC stability in future climates. Currently, the global warming threshold for an AMOC collapse is estimated to be 4 °C (Armstrong-McKay et al., 2022). In the future, the carbon content of the ocean-atmosphere system will increase, potentially increasing the MEW which can change the likelihood of a bifurcation induced AMOC collapse. In this study we focused on slow, bifurcation induced tipping of the AMOC, while the AMOC is also able to tip due to faster processes (e.g. density changes related to temperature variations) resulting in noise-induced tipping (Castellana et al., 2019; Jacques-Dumas et al., 2023; van Westen et al., 2024), and due to rate-induced tipping (Alkhayuon et al., 2019; Lohmann and Ditlevsen, 2021). The mechanisms presented here might influence these noise-induced transitions as well. We hope this work inspires further research on the dependency of the AMOC MEW on the carbon cycle in more detailed models, to further investigate the relevance of the mechanism found in this study, and provide a better quantification for the influence of the marine carbon cycle on the MEW of the AMOC.

*Code and data availability.* All model code, data and scripts are available at https://doi.org/10.5281/zenodo.10005999 (Boot et al., 2023a). AUTO-07p can be downloaded from https://github.com/auto-07p/auto-07p (Doedel, E J and Paffenroth, R C and Champneys, A C and Fairgrieve, T F and Kuznetsov, Yu A and Oldeman, B E and Sandstede, B and Wang, X J, 2021).

## Appendix A: Additional couplings, feedbacks and simulations

Besides the couplings and feedbacks presented in the main text we have introduced one additional coupling and two additional feedbacks to the carbon cycle. A summary of these cases and the results can be seen in Table A1 and Fig. A2. The main effects of these additional coupling and feedbacks is a shift in atmospheric $pCO_2$ values on the on-branch for cases with the piston velocity feedback (Eq. A3 and Eq. A4). This shift is larger when also the climate sensitivity feedback is used. A description of the additional coupling and feedbacks is given below.

The additional coupling we have introduced is the addition of dilution fluxes for both DIC and Alk related to the freshwater fluxes $E_s$ and $E_a$ (Eq. A1). Increasing the concentrations of DIC and Alk due to evaporation and decreasing the concentrations due to a net influx of freshwater at the surface.

$$C_{dil,i} = \lambda_D \times (E_s + E_a) \times \frac{C_i}{V_i} \tag{A1}$$

Where $C_i$ is the tracer concentration in box i and $V_i$ the volume, and $\lambda_D$ is a parameter that determines whether the coupling
is used ($\lambda_D = 1$) or not ($\lambda_D = 0$). The dilutive fluxes for Alk are modelled in a similar fashion.

A first additional feedback we introduce is a linear temperature dependency in the biological efficiency (Eq. A2) which was introduced in the biological coupling. Under an SST increase, the efficiency will decrease following

$$\epsilon_i = (\lambda_\epsilon \times -0.1\Delta T) + \epsilon_{i,base} \tag{A2}$$

For this feedback it is necessary to also use the climate sensitivity feedback and the strength can be regulated with $\lambda_\epsilon$.
The second additional feedback allows the piston velocity ($k_w$) to vary with the SSTs (Eq. A3). When the climate sensitivity feedback is used, this also affects the piston velocity. The temperature dependency is introduced by making the piston velocity a function of the Schmidt number (Eq. A4) following

$$k_{w,i} = (1 - \lambda_P) \times k_{w,ibase} + \lambda_P k_{w,ibase} \times (\frac{Sc_i}{660})^{-0.5} \tag{A3}$$

Where

$$Sc_i = 2116.8 - 136.25T_i + 4.7353T_i^2 - 0.092307T_i^3 + 0.0007555T_i^4 \tag{A4}$$

In this case the feedback can either be switched on ($\lambda_P = 1$) or off ($\lambda_P = 0$). Without this feedback the piston velocity is similar for all boxes, but with this feedback the piston velocity will differ per box.

**Table A1.** Additional cases not included in the main text using additional feedbacks as described in this document. Results of these cases can be seen in Fig. A2.

| Notation | S-1 | S-2 | S-3 | S-4 | S-5 | S-6 | S-7 | S-8 | S-9 | S-10 |
|----------|-----|-----|-----|-----|-----|-----|-----|-----|-----|------|
| $\lambda_{BI}$ | 1 | 1 | 1 | 1 | 1 | 1 | 1 | 1 | 1 | 1 |
| $\lambda_T$ | 1 | 0 | 0 | 1 | 1 | 1 | 0 | 0 | 1 | 1 |
| $\lambda_P$ | 0 | 0 | 1 | 1 | 1 | 0 | 0 | 1 | 1 | 1 |
| $\lambda_D$ | 0 | 1 | 0 | 0 | 0 | 0 | 1 | 0 | 0 | 0 |
| $\lambda_\epsilon$ | 0 | 0 | 0 | 0 | 1 | 0 | 0 | 0 | 0 | 1 |
| $\lambda_E$ | 0 | 0 | 0 | 0 | 0 | 1 | 1 | 1 | 1 | 1 |

## Appendix B:  Model parameters

The model parameters are presented in Tables B3 to B5.

## Appendix C:  Model equations

There are in total 30 state variables: salinity, DIC, alkalinity, and $PO_4$ in the 7 boxes, the pycnocline depth D, and atmospheric $pCO_2$. The state variables in the deep Atlantic box are determined using conservation laws. The salinity equations are given by Eq. C1-C6, the conservation of salt in the model is given by Eq. C8, and the pycnocline depth is determined using Eq. C7. The volume fluxes are determined using Eq. C9 to C13, and the equation of state is given by Eq. C14. The equations for the carbon cycle model are given by Eq. C15 to Eq. C27.

$$\frac{d(V_t S_t)}{dt} = q_S(\theta(q_S)S_{ts} + \theta(-q_S)S_t + q_U S_d - \theta(q_N)q_N S_t + r_s(S_{ts} - S_t) + r_N(S_n - S_t) + 2E_s S_0 \tag{C1}$$

$$\frac{d(V_{ts} S_{ts})}{dt} = q_{Ek}S_s - q_e S_{ts} - q_S(\theta(q_S)S_{ts} + \theta(-q_S)S_t) + r_S(S_t - S_{ts}) \tag{C2}$$

$$V_n \frac{dS_n}{dt} = \theta(q_N)q_N(S_t - S_n) + r_N(S_t - S_n) - (E_s + E_a)S_0 \tag{C3}$$

$$V_s \frac{dS_s}{dt} = q_S(\theta(q_S)S_d + \theta(-q_S)S_s) + q_e S_{ts} - q_{Ek}S_s - (E_p + E_s - E_a)S_0 + (r_P + \psi_1)(S_{ps} - S_s) \tag{C4}$$

$$V_{ps} \frac{dS_{ps}}{dt} = (\gamma_1 + \psi_1) * (S_{pd} - S_{ps}) + (r_P * (S_s - S_{ps})) + E_p \tag{C5}$$

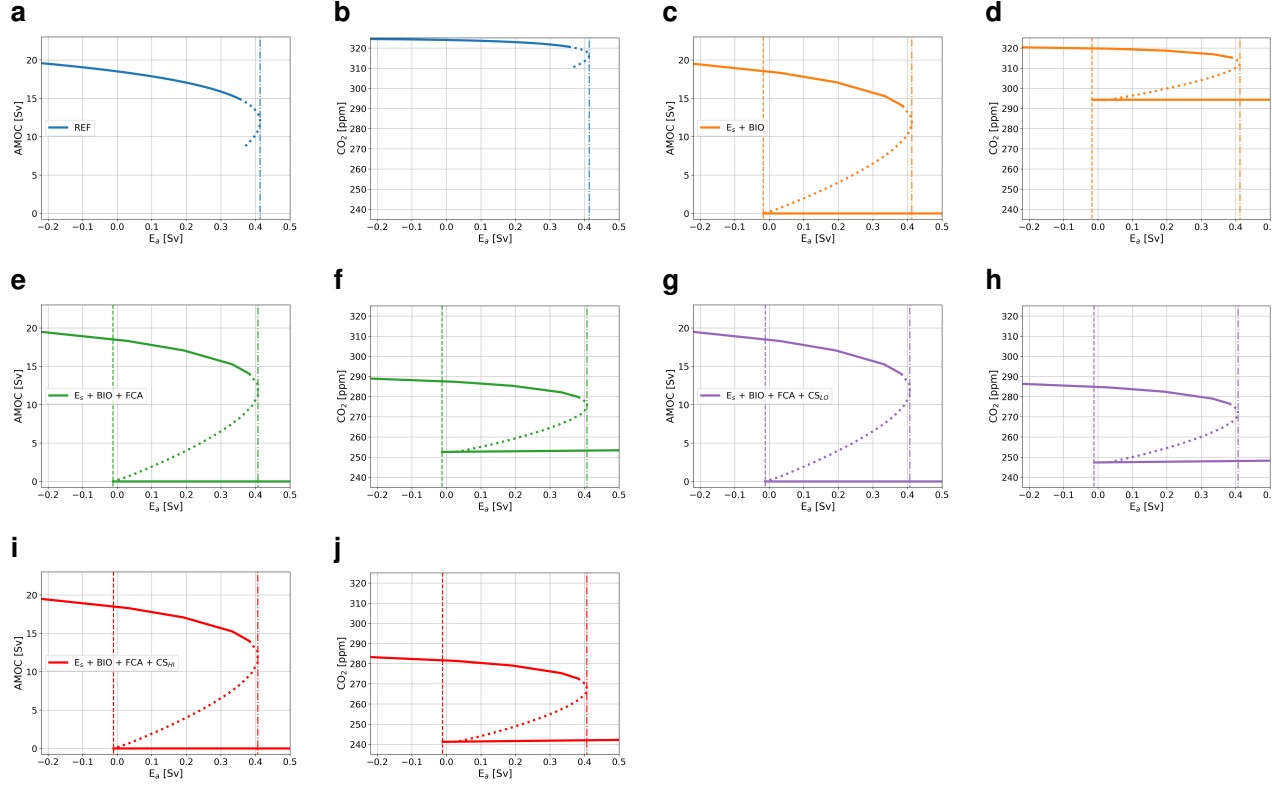

**Figure A1.** As Fig. 3 but for the other cases discussed in the main text. (a, b) the case without additional coupling (REF) where the off state cannot be simulated. (c, d) the case with the CMIP6 based $E_s$ and biological coupling ($E_s$ + BIO). (e, f) the case where also the rain ratio feedback is applied ($E_s$ + BIO + FCA). (g-j) as (e, f) but also with the climate sensitivity feedback, with a low sensitivity (g, h; $E_s$ + BIO + FCA + $CS_{LO}$) and a high sensitivity (i, j; $E_s$ + BIO + FCA + $CS_{HI}$). (a, c, e, g, i) are the AMOC strength in Sv versus $E_a$ in Sv, and (b, d, f, h, j) are the $CO_2$ concentration in the atmosphere in ppm versus $E_a$ in Sv.

$$V_{pd}\frac{dS_{pd}}{dt} = \gamma_1 * (S_{ps} - S_{pd}) + \psi_1(S_d - S_{pd}) \tag{C6}$$

$$(A + \frac{L_{xA}L_y}{2})\frac{dD}{dt} = q_U + q_{Ek} - q_e - \theta(q_N)q_N \tag{C7}$$

$$S_0V_0 = V_nS_n + V_dS_d + V_tS_t + V_{ts}S_{ts} + V_sS_s + V_{ps}S_{ps} + V_{pd} + S_{pd} \tag{C8}$$

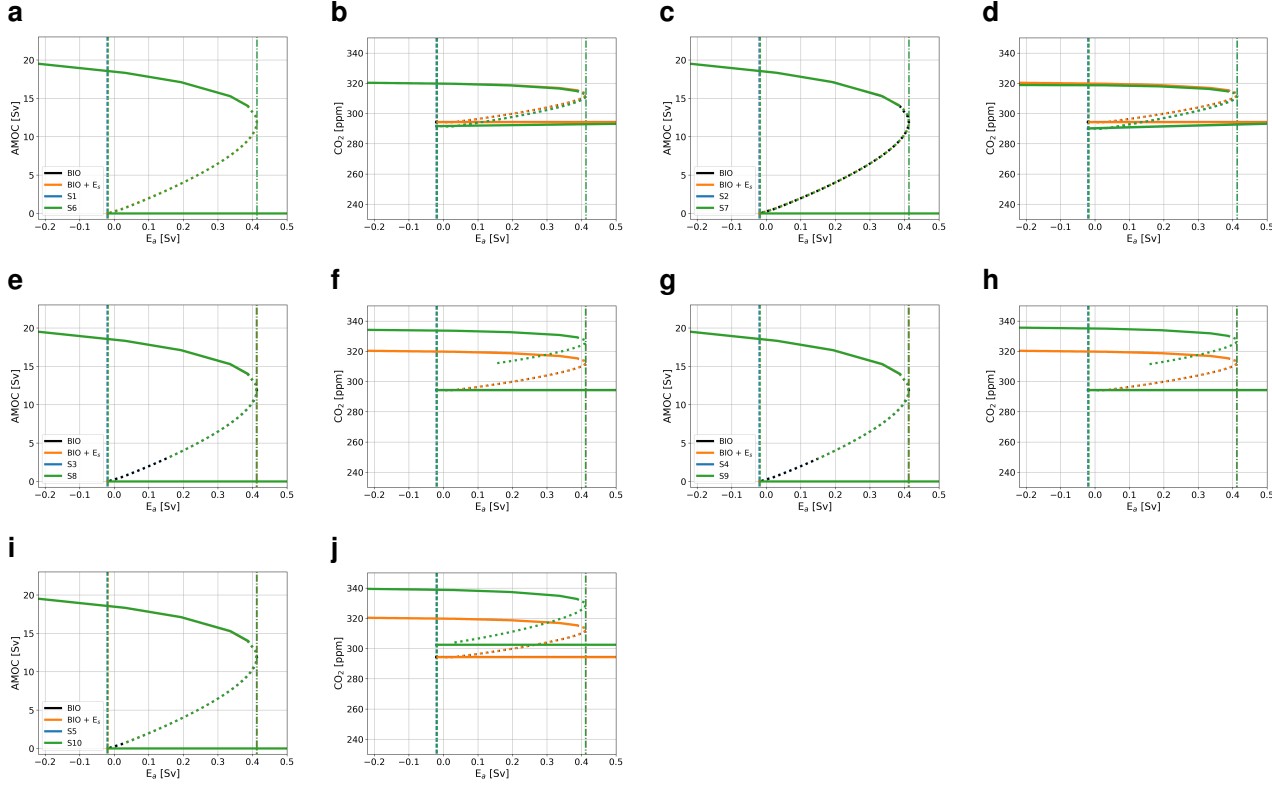

**Figure A2.** Bifurcation diagrams showing the sensitivity of the model to $E_a$ for additional cases as defined in Table A1. Solid lines represent stable steady state solutions, dotted lines represent unstable states, dash-dotted lines represent the location of the saddle-node bifurcation on the on-branch, and dashed lines the location of the saddle-node bifurcation on the off-branch. The black lines represent a case with only the biological coupling (BIO), the orange lines with the logarithmic CMIP6 based $E_s$ and biological coupling ($E_s$ + BIO), and the blue and green lines represent the cases defined in Table A1. Results are for the AMOC strength in Sv (a, c, e, g, i) and atmospheric $pCO_2$ in ppm (b, d, f, h, j).

Where $\theta$ is a step function which takes a value of 1 for a positive argument, and takes a value of 0 for a negative argument.
The volume fluxes are given by:

$$q_{Ek} = \frac{\tau L_{xS}}{\rho_0 |f_S|} \tag{C9}$$

$$q_e = A_{GM} \frac{L_{xA}}{L_y} D \tag{C10}$$

**Table B1.** Symbol (column 1), description (column 2), value (column 3), and units (column 4) of the general parameters used in the ocean circulation model based on Cimatoribus et al. (2014).

| Symbol | Description | Value | Units |
|---|---|---|---|
| $V_{0,A}$ | Total volume of the Atlantic basin | $3 \times 10^{17}$ | $\mathrm{m}^3$ |
| $V_n$ | Volume of box $n$ | $3 \times 10^{15}$ | $\mathrm{m}^3$ |
| $V_s$ | Volume of box $s$ | $9 \times 10^{15}$ | $\mathrm{m}^3$ |
| $A_t$ | Surface area box $t$ | $1 \times 10^{14}$ | $\mathrm{m}^2$ |
| $L_{xA}$ | Zonal extent of the Atlantic Ocean at its southern end | $1 \times 10^7$ | m |
| $L_y$ | Meridional extent of the frontal region of the Southern Ocean | $1 \times 10^6$ | m |
| $L_{xS}$ | Zonal extent of the Southern Ocean | $3 \times 10^7$ | m |
| $\tau$ | Average zonal wind stress amplitude | 0.1 | $\mathrm{N\,m}^{-2}$ |
| $A_{GM}$ | Eddy diffusivity | 1700 | $\mathrm{m}^2\,\mathrm{s}^{-1}$ |
| $f_S$ | Coriolis parameter | $-1 \times 10^{-4}$ | $\mathrm{s}^{-1}$ |
| $\rho_0$ | Reference density | 1027.5 | $\mathrm{kg\,m}^{-3}$ |
| $\kappa$ | Vertical diffusivity | $1 \times 10^{-5}$ | $\mathrm{m}^2\,\mathrm{s}^{-1}$ |
| $S_0$ | Reference salinity | 35 | g/kg |
| $T_0$ | Reference temperature | 5 | °C |
| $T_{n,base}$ | Base temperature box $n$ | 5 | °C |
| $T_{ts,base}$ | Base temperature box $ts$ | 10 | °C |
| $\eta$ | Hydraulic constant | $3 \times 10^4$ | $\mathrm{m\,s}^{-1}$ |
| $\alpha$ | Thermal expansion coefficient | $2 \times 10^{-4}$ | $\mathrm{K}^{-1}$ |
| $\beta$ | Haline contraction coefficient | $8 \times 10^{-4}$ | $(\mathrm{g/kg})^{-1}$ |
| $r_S$ | Transport by the southern subtropical gyre | $10 \times 10^6$ | $\mathrm{m}^3\,\mathrm{s}^{-1}$ |
| $r_N$ | Transport by the northern subtropical gyre | $5 \times 10^6$ | $\mathrm{m}^3\,\mathrm{s}^{-1}$ |

$$q_U = \frac{\kappa A}{D} \tag{C11}$$

$$q_N = \eta \frac{\rho_n - \rho_{ts}}{\rho_0} D^2 \tag{C12}$$

$$q_S = q_{Ek} - q_e \tag{C13}$$

$$\rho_i = \rho_0 (1 - \alpha(T_i - T_0) + \beta(S_i - S_0)) \tag{C14}$$

**Table B2.** Symbol (column 1), description (column 2), value (column 3), and units (column 4) of the general parameters used in the ocean circulation model added or changed with respect to Cimatoribus et al. (2014)

| Symbol | Description | Value | Units |
|--------|-------------|-------|-------|
| $E_s$ | Symmetric freshwater flux | $0.39 \times 10^6$ | $\text{m}^3 \text{ s}^{-1}$ |
| $E_p$ | Freshwater flux from box $ps$ to box $s$ | $0.99 \times 10^6$ | $\text{m}^3 \text{ s}^{-1}$ |
| $V_0$ | Total volume of the ocean | $1.5 \times 10^{18}$ | $\text{m}^3$ |
| $V_{ps}$ | Volume Box $ps$ | $9 \times 10^{16}$ | $\text{m}^3$ |
| $V_{pd}$ | Volume Box $pd$ | $1.11 \times 10^{18}$ | $\text{m}^3$ |
| $d_{ps}$ | Depth Box $ps$ | 300 | m |
| $d_{fn}$ | Floor depth Box $n$ | 300 | m |
| $d_{ft}$ | Floor depth Box $t$ | variable ($D$) | m |
| $d_{fts}$ | Floor depth Box $ts$ | variable ($D$) | m |
| $d_{fs}$ | Floor depth Box $s$ | 300 | m |
| $d_{fd}$ | Floor depth Box $d$ | 4000 | m |
| $T_{t,base}$ | Base temperature Box $t$ | 23.44 | °C |
| $T_{s,base}$ | Base temperature Box $s$ | 0.93 | °C |
| $T_d$ | Temperature Box $d$ | 1.8 | °C |
| $T_{ps}$ | Temperature Box $ps$ | 23.44 | °C |
| $T_{pd}$ | Temperature Box $pd$ | 1.8 | °C |
| $r_P$ | Transport by the subtropical gyre between box $s$ and $ps$ | $90 \times 10^6$ | $\text{m}^3 \text{ s}^{-1}$ |

Where i represents any box.

The carbon cycle equations are given by Eq. C15 to Eq. C19. The different fluxes are determined using Eq. C20 to Eq. C27.

$$\frac{d[DIC]_i}{dt} = C_{phys,i} + C_{bio,i} + C_{carb,i} + C_{air,i} + C_{river,t} \tag{C15}$$

$$\frac{d[Alk]_i}{dt} = A_{phys,i} + A_{carb,i} + A_{river,t} \tag{C16}$$

$$\frac{d[PO_4^{3-}]_i}{dt} = P_{phys,i} + P_{bio,i} + P_{river,t} \tag{C17}$$

$$\frac{dC_{tot}}{dt} = C_{river,t} \times V_t + \sum_{i=1}^{5}(C_{carb,i}V_i) + \sum_{i=1}^{5}(C_{bio,i}V_i) \tag{C18}$$

**Table B3.** Symbol (column 1), description (column 2), value (column 3), and units (column 4) of the general parameters used in the carbon cycle model based on Boot et al. (2022).

| Symbol | Description | Value | Units |
|---|---|---|---|
| $V_{at}$ | Volume of the atmosphere | $1.76 \times 10^{20}$ | $m^3$ |
| $\psi_1$ | Global overturning circulation | $18 \times 10^6$ | $m^3\ s^{-1}$ |
| $\gamma_1$ | Bidirectional mixing term between box $ps$ and $pd$ | $30 \times 10^6$ | $m^3\ s^{-1}$ |
| $n$ | Order of $CaCO_3$ dissolution kinetics | 1 | - |
| $P_C$ | Mass percentage of C in $CaCO_3$ | 0.12 | - |
| $D_{Ca}$ | Constant dissolution rate of $CaCO_3$ | $2.75 \times 10^{-13}$ | $mol\ m^{-3}\ s^{-1}$ |
| $W_{SC}$ | Constant silicate weathering | $2.4 \times 10^{-12}$ | $mol\ m^{-3}\ s^{-1}$ |
| $W_{SV}$ | Variable silicate weathering parameter | $1.6 \times 10^{-8}$ | $mol\ m^{-3}\ atm^{-1}\ s^{-1}$ |
| $W_{CV}$ | Variable carbonate weathering parameter | $6.3 \times 10^{-8}$ | $mol\ m^{-3}\ atm^{-1}\ s^{-1}$ |
| $k_{Ca}$ | Constant $CaCO_3$ dissolution rate | $4.4 \times 10^{-6}$ | $s^{-1}$ |
| $b$ | Exponent in Martin's law | 0.75 | - |
| $d_0$ | Reference depth for biological productivity | 100 | m |
| $k_{w,base}$ | Base piston velocity | 3 | m/day |
| $R_{C:P}$ | Redfield C:P ratio | 130 | mol C/mol P |
| $R_{P:C}$ | Redfield P:C ratio | 1/130 | mol P/mol C |
| $[Ca]_n$ | Calcium concentration Box $n$ | $0.01028 \times S_n$ | $mol\ m^{-3}$ |
| $[Ca]_t$ | Calcium concentration Box $t$ | $0.01028 \times S_t$ | $mol\ m^{-3}$ |
| $[Ca]_{ts}$ | Calcium concentration Box $ts$ | $0.01028 \times S_{ts}$ | $mol\ m^{-3}$ |
| $[Ca]_s$ | Calcium concentration Box $s$ | $0.01028 \times S_s$ | $mol\ m^{-3}$ |
| $[Ca]_d$ | Calcium concentration Box $d$ | $0.01028 \times S_d$ | $mol\ m^{-3}$ |

$$\frac{dAlk_{tot}}{dt} = Alk_{river,t} \times V_t + Alk_{river,ps} \times V_{ps} + \sum_{i=1}^{7}(Alk_{carb,i}V_i) \tag{C19}$$

In these equations the different terms represent advective fluxes ($X_{phys}$), biological fluxes ($X_{bio}$), carbonate fluxes ($X_{carb}$), air-sea gas exchange ($C_{air}$) and the river influx ($X_{river}$). From these fluxes, $C_{air}$ only acts on the surface boxes, and $X_{river}$ only on box $t$ and box $ps$. $X_{phys}$ is determined following:

$$X_{phys,i} = \frac{1}{V_i}(\sum_{i=1}(q_{j \to i} \times X_j) - \sum_{i=1}(q_{i \to j} \times X_i)) \tag{C20}$$

This equation represents that the concentration of tracer X changes through an advective flux flowing out of box i to box j ($q_{i \to j}$ times the concentration in box i ($X_i$), and a flux flowing into box i from box j ($q_{j \to i}$) times the concentration in box j ($X_j$). There can be fluxes from multiple boxes into one box.

**Table B4.** Symbol (column 1), description (column 2), value (column 3), and units (column 4) of the parameters used in the carbon cycle model that have been changed compared to Boot et al. (2022).

| Symbol | Description | Value | Units |
|---|---|---|---|
| $Z_{n,base}$ | Base biological production Box $n$ | 1.9 | mol C m$^{-2}$ yr$^{-1}$ |
| $Z_{t,base}$ | Base biological production Box $t$ | 2.1 | mol C m$^{-2}$ yr$^{-1}$ |
| $Z_{ts,base}$ | Base biological production Box $ts$ | 2.1 | mol C m$^{-2}$ yr$^{-1}$ |
| $Z_{s,base}$ | Base biological production Box $s$ | 1.1 | mol C m$^{-2}$ yr$^{-1}$ |
| $\epsilon_{n,base}$ | Base biological efficiency Box $n$ | 0.1 | - |
| $\epsilon_{t,base}$ | Base biological efficiency Box $t$ | 0.5 | - |
| $\epsilon_{ts,base}$ | Base biological efficiency Box $ts$ | 0.3 | - |
| $\epsilon_{s,base}$ | Base biological efficiency Box $s$ | 0.1 | - |
| $F_{Ca,base}$ | Base rain ratio | 0.15 | - |
| $pCO_{2,0}$ | Base atmospheric pCO$_2$ value | 320 | ppm |

**Table B5.** The symbols and description of the equilibrium constants are presented in the first two columns. The third column presents the source of the used expression.

| Symbol | Description | Expression |
|---|---|---|
| $K_0$ | Solubility constant | Weiss (1974) |
| $K_1$ | First dissociation constant of carbonic acid | Lueker et al. (2000) |
| $K_2$ | Second dissociation constant of carbonic acid | Lueker et al. (2000) |
| $K_{sp,base}$ | Equilibrium constant for CaCO$_3$ dissolution | Mucci (1983) |
| $K_{sp,press}$ | Pressure correction for $K_{sp,base}$ | Millero (1983) |

$$C_{air.i} = \frac{K_{0,i} \times k_{w,i} \times \rho_0 \times (CO_2^{atm} - pCO_{2,i})}{V_i} \tag{C21}$$

For i is *n, t, ts, s* or *ps*. $K_0$ is the solubility constant, $k_w$ the piston velocity, $CO_2^{atm}$ the atmospheric CO$_2$ concentration, pCO$_2$ the partial pressure of CO$_2$ in the ocean and V the volume of the ocean box.

$$C_{carb.i} = -\frac{Z_i \times A_i \times F_{Ca,i}}{V_i} + ([CO_3^{2-}]_i [Ca^{2+}]_i) \rho_0 k_{Ca} (1 - \frac{([CO_3^{2-}]_i [Ca^{2+}]_i)}{K_{sp,i}})^n \times PerC + DC \tag{C22}$$

For i is *n, t, ts, s* or *ps*. $Z$ represent biological production, $A$ the surface area of the box, $F_{Ca}$ the rain ratio and $V$ the volume. Other variables are the carbonate ion concentration ($[CO_3^{2-}]$), calcium concentration ($[Ca^{2+}]$), and equilibrium constant for CaCO$_3$ dissolution ($K_{sp}$).

For box *pd* the carbonate flux is determined following

$$C_{carb.i} = ([CO_3^{2-}]_{pd}[Ca^{2+}]_{pd})\rho_0 k_{Ca}(1 - \frac{([CO_3^{2-}]_{pd}[Ca^{2+}]_{pd})}{K_{sp,pd}})^n \times PerC + ([CO_3^{2-}]_{pd}[Ca^{2+}]_{pd})\rho_0 k_{Ca} \times$$

$$(1 - \frac{([CO_3^{2-}]_{pd}[Ca^{2+}]_{pd})}{K_{sp,sed}})^n \times PerC + DC \tag{C23}$$

Where there is a distinction between water column dissolution of $CaCO_3$ and dissolution in the sediments.

The biological fluxes in the surface ocean are given by:

$$C_{bio,i} = \frac{Z_i \times A_i}{V_i} \times (\frac{d_{fi}}{d_0})^{-b} \tag{C24}$$

For i is *n, t, ts, s* or *ps*. $Z$ represent biological production, $A$ the surface area of the box, $V$ the volume, and $d_{fi}$ the floor depth of the box.

The biological flux for box *pd* is given by:

$$C_{bio,i} = \frac{Z_ps \times A_ps}{V_ps} \times ((\frac{d_{fps}}{d_0})^{-b} - (\frac{d_{tot}}{d_0})^{-b}) \tag{C25}$$

Alkalinity and phosphate fluxes are proportionate to DIC fluxes following:

$$A_{carb.i} = 2 \times C_{carb.i} \tag{C26}$$

$$P_{bio,i} = r_{P:C} \times C_{bio,i} \tag{C27}$$

Where $r_{P:C}$ is a constant stoichiometric P to C parameter.

An explanation and the value of all parameters are given in the tables in Appendix B.

*Author contributions.* A.A.B. constructed the AUTO version of the coupled model and obtained and analyzed the results. All authors contributed to the writing of the paper.

*Competing interests.* The authors declare that they have no conflict of interest.

*Financial support.* This research has been supported by the Netherlands Earth System Science Centre (grant no. 024.002.001). The work of A.A.B. and H.A.D. was also funded by the European Research Council through the ERC-AdG project TAOC (PI: Dijkstra, project 101055096). The work of A.S.vdH was also funded by the Dutch Research Council (NWO) through the NWO-Vici project 'Interacting climate tipping elements: When does tipping cause tipping?' (project VI.C.202.081)

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

**Table A1.** List of CMIP6 models used in this study: model name (column 1), member used (column 2), corresponding variable (column 3; either water flux (wfo) or virtual salt flux (vsf)), reference (column 4).

| Name | Member | Variable | Reference |
|---|---|---|---|
| ACCESS-CM2 | r1i1p1f1 | wfo | Dix et al. (2019) |
| ACCESS-ESM1-5 | r1i1p1f1 | wfo | Ziehn et al. (2019) |
| CESM2 | r1i1p1f1 | vsf | Danabasoglu (2019) |
| CESM2-WACCM-FV2 | r1i1p1f1 | vsf | Danabasoglu (2020) |
| CMCC-CM2-HR4 | r1i1p1f1 | wfo | Scoccimarro et al. (2021) |
| CMCC-ESM2 | r1i1p1f1 | wfo | Lovato et al. (2021) |
| CNRM-CM6-1-HR | r1i1p1f2 | wfo | Voldoire (2019) |
| CNRM-ESM2-1 | r1i1p1f2 | wfo | Seferian (2018) |
| CanESM5 | r1i1p1f1 | wfo | Swart et al. (2019a) |
| CanESM5-1 | r1i1p1f1 | wfo | Swart et al. (2019b) |
| E3SM-2-0 | r1i1p1f1 | wfo | e3s (2022) |
| E3SM-2-0-NARRM | r1i1p1f1 | wfo | e3s (2023) |
| FGOALS-f3-L | r1i1p1f1 | vsf | Yu (2019) |
| FGOALS-g3 | r2i1p1f1 | vsf | Li (2019) |
| FIO-ESM-2-0 | r1i1p1f1 | wfo | Song et al. (2020) |
| GFDL-CM4 | r1i1p1f1 | wfo | Guo et al. (2018) |
| GFDL-ESM4 | r1i1p1f1 | wfo | Krasting et al. (2018) |
| GISS-E2-1-G | r1i1p1f1 | wfo | for Space Studies (NASA/GISS) (2018) |
| GISS-E2-2-G | r1i1p1f1 | wfo | for Space Studies (NASA/GISS) (2019) |
| HadGEM3-GC31-LL | r1i1p1f3 | wfo | Ridley et al. (2019) |
| HadGEM3-GC31-MM | r1i1p1f3 | wfo | Ridley et al. (2020) |
| IPSL-CM5A2-INCA | r1i1p1f1 | wfo | Boucher et al. (2020) |
| IPSL-CM6A-LR | r1i1p1f1 | wfo | Boucher et al. (2018) |
| MCM-UA-1-0 | r1i1p1f1 | wfo | Stouffer (2019) |
| MIROC-ES2L | r1i1p1f2 | wfo | Hajima et al. (2019) |
| MIROC6 | r1i1p1f1 | wfo | Tatebe and Watanabe (2018) |
| MPI-ESM-1-2-HAM | r1i1p1f1 | wfo | Neubauer et al. (2019) |
| MPI-ESM1-2-LR | r1i1p1f1 | wfo | Wieners et al. (2019) |
| MRI-ESM2-0 | r1i1p1f1 | wfo | Yukimoto et al. (2019) |
| NESM3 | r1i1p1f1 | wfo | Cao and Wang (2019) |
| NorCPM1 | r1i1p1f1 | vsf | Bethke et al. (2019) |
| NorESM2-MM | r1i1p1f1 | vsf | Bentsen et al. (2019) |
| SAM0-UNICON | r1i1p1f1 | wfo | Park and Shin (2019) |