# Peer review of "Potential effect of the marine carbon cycle on the multiple equilibria window of the Atlantic Meridional Overturning Circulation"

_Earth System Dynamics, 2023_

## Author Comment (AC1)

**MS-No.:** ESD-2023-30

**Title:** Potential effect of the marine carbon cycle on the multiple equilibria window of the Atlantic Meridional Overturning Circulation

**Authors:** Amber A. Boot, Anna S. von der Heydt and Henk A. Dijkstra

**Point-by-point reply to reviewer #1**

June 7, 2024

We thank the reviewer for their careful reading and for the useful comments on the manuscript.

**Summary:**
*Boot et al. study the "multiple equilibria window" (MEW) of the Atlantic Meridional Overturning Circulation (AMOC) in a coupled ocean-carbon cycle box model. Specifically, they study the interactions between the AMOC and the carbon cycle, and show, among other things, that:*

1. *AMOC off states have lower atmospheric $pCO_2$.*

2. *the MEW widens when more carbon is added to the system.*

*I think that this is an exciting paper. The model seems reasonable (and indeed is based on multiple previous studies) and the methodological approach is sound. The AMOC and its nonlinear behaviour are subjects of great relevance, both with respect to present or future tipping risk as well as for understanding paleoclimate dynamics, and the paper derives some interesting new insights. I always appreciate seeing dynamical systems approaches (and indeed, AUTO) used for these purposes. However, I do think that the paper's key insights are still a little obscured behind the modelling details, and I recommend a few revisions to bring them out more clearly.*

**Specific comments:**

1. *First, it's challenging on an initial read to keep track of all of the cases and what they mean. This is challenging to get around without major rewrites (which I do not think are needed), but I think at minimum*

*Table 1 could be made more helpful by describing clearly what all the lambda values represent. This could be done either within the table itself, or perhaps more productively in the caption.*

**Author's reply:**
We agree that it can be difficult to keep track of all the different cases.

**Changes in manuscript:**
We will increase the clarity of the caption for Table 1 where we will more elaborately explain what the different lambda values represent.

2. *Next, Figure 1: I think it's worth mentioning explicitly in the caption that the strength of the AMOC downwelling is set by the meridional density gradient between ts and n. Understanding exactly how AMOC strength is set in the model will help readers later on when mechanisms are explained.*

**Author's reply:**
We agree.

**Changes in manuscript:**
Suggestion followed.

3. *Figure 3: I found this quite confusing at first read, not least because of the overlap between many of the curves. If the key point of this figure is to show the general shape of the AMOC bifurcation diagram as well as to illustrate that off states have lower pCO2, perhaps it might be worth showing only this: i.e. AMOC vs Ea and pCO2 vs Ea for one single case (and moving the other cases to the Appendix). This is not essential, but I offer it as a suggestion.*

**Author's reply:**
We agree that the figure can be confusing.

**Changes in manuscript:**
Suggestion followed.

4. *Figure 4: my first comment is that this is really big compared to other figures that strike me as equally important, e.g. 5a. Second, it seems like what really matters are not the blue and orange lines themselves but the spaces they demarcate – why not label them accordingly? e.g. the region between the lines is precisely the MEW, the region above the blue line is one where only the off state is stable, and the region below the orange line is that where only the on-state is stable. Third, why not include CO2 levels as a second x-axis at the top of the graph which maps nonlinearly onto Es? I think these changes would make the figure vastly easier to understand at first glance.*

**Author's reply:**
Thank you for the useful suggestions.

**Changes in manuscript:**
Suggestions followed: we will label the monostable regimes and multiple equilibria window explicitly in the figure and use a second x-axis for the $CO_2$ concentrations instead of the green line and decrease the size of the figure.

5. *Figure 5: My main comment here is that this could be much larger. For example, it seems like 5a shows a major result of the paper, but it's small and hard to read. Maybe a and b could be on the top row and c in the middle on the bottom row? Also, it's worth mentioning in the caption the result from Caves et al. (2016) that total carbon content has varied between 24,000 and 96,000 Pg C, to make the reader understand immediately that the changes explored in the figure are reasonable.*

**Author's reply:**
Thank you for the suggestions, and indeed Fig. 5a shows the main result of this paper.

**Changes in manuscript:**
All suggestions followed.

6. *Figure 6. I guess this is probably a Latex quirk, but it's strange to me that it's placed after the Appendix and all of the references – this makes it easy to miss at first glance. It would be good to place it much more prominently near the end of the text. Finally, I suggest replacing dTC/dt with d[DIC]/dt (if indeed that's what's meant).*

   **Author's reply:**
   It is indeed a Latex quirk. We have chosen not to use DIC since atmospheric $pCO_2$ is also part of the total carbon (TC) content of the system.

   **Changes in manuscript:**
   We will make sure that Fig. 6 will placed correctly in the main text. We will clarify in the caption of the figure that TC represents DIC and atmospheric $pCO_2$.

**More minor comments:**

1. *Line 20: It may be worth mentioning studies reporting a present-day AMOC weakening, e.g. Caesar et al. (2018), Boers (2021), Ditlevsen and Ditlevsen (2023).*

   **Author's reply:**
   We agree.

   **Changes in manuscript:**
   We will mention papers studying AMOC collapses in present-day climate as suggested by the reviewer.

2. *Lines 37-38: I'm not directly familiar with the studies by Barker et al., but at a glance it seems like these are primarily observational (i.e.*

*not model-based). It may be worth mentioning this, as it highlights the novelty of the authors' work.*

**Author's reply:**
The work of Barker et al. is indeed observation based.

**Changes in manuscript:**
We will mention this in the revision.

3. *Line 38: "of how"?*

   **Author's reply:**
   -

   **Changes in manuscript:**
   Suggestion followed.

4. *Line 54: "eddy-induced" (consistent with wind-induced)*

   **Author's reply:**
   -

   **Changes in manuscript:**
   Suggestion followed.

5. *Line 87: "to form the model used..."*

   **Author's reply:**
   -

   **Changes in manuscript:**
   Suggestion followed.

6. *Line 97: I suggest always using "riverine flux" instead of "river flux" for clarity; "river flux" is repeated a number of times throughout the paper.*

   **Author's reply:**
   We agree.

   **Changes in manuscript:**
   Suggestion followed.

7. *Line 106/Eq. (1): It seems like there is a sum over all j missing here?*

   **Author's reply:**
   You are right. Formally there should indeed be a sum over j there.

   **Changes in manuscript:**
   Suggestion followed.

8. *Line 128/Eq. (4): do you have some more justification for this? e.g. the 0.81 power law?*

   **Author's reply:**
   This value is taken from Ridgwell et al. (2007) which represents thermodynamic calcification rates. They use this 0.81 value as a calibration parameter in the GENIE-1 Earth System Model.

   **Changes in manuscript:**
   We will cite the source of the function (i.e. Ridgwell et al. (2007)), and provide more background on the power law. We will also fix a typo in this equation, i.e. the expression between the brackets should be $(\frac{[Ca^{2+}][CO_3^{2-}]}{K_{sp,i}} - 1)$

9. *Line 151: Eq. (6): The linear dependence on atmospheric CO2 here (e.g. as opposed to other powers) is a fairly strong assumption that*

*should probably be discussed.*

**Author's reply:**
The linear dependence used here comes from the original SCP-M (O'Neill et al., 2019) which is based on the works of Toggweiler (2008). In models, such as LOSCAR (Zeebe, 2012), a model of similar complexity designed to simulate the long term carbon cycle, where a power law is used. Specifically in LOSCAR the power law causes atmospheric $pCO_2$ to converge in time to a predefined $pCO_2$ value. Since we apply a steady state approach this method can not be used. There are also models such as COPSE (Bergman et al., 2004) and GEOCARB-SULF (Royer, 2014) that use a much more complex weathering term including effects of temperature (which is linked to atmospheric $pCO_2$) and vegetation. This type of parameterization is too complex for our model.

Other powers could obviously be used in the model. Powers larger than one will decrease the sensitivity of the model to changes in the burial of $CaCO_3$ in the ocean, and powers smaller than one will increase the sensitivity of the model. Given that the model does not seem to be very sensitive to non-linear feedbacks in the carbon cycle, we would not expect additional non-linear behavior.

**Changes in manuscript:**
We will add a few lines in the discussion where we highlight that the parameterization we use is linear and based on previous work. We will also what it would mean for the results, as described above, when a nonlinear dependence is used.

10. *Line 189: (Andersson et al. 2017)*

    **Author's reply:**
    -

    **Changes in manuscript:**
    Suggestion followed.

11. *Line 233: I think the usage of "saddle nodes" is confusing, and recommend that every instance of this be replaced with "saddle-node bifurcations".*

    **Author's reply:**
    We agree.

    **Changes in manuscript:**
    Suggestion followed.

12. *Figure 4: which case are these results from?*

    **Author's reply:**
    They are from the uncoupled case, i.e. without active carbon cycle model in there.

    **Changes in manuscript:**
    We will clarify this in the caption.

13. *Line 325: and rate-induced tipping, see e.g. Alkhayuon et al. (2019), Lohmann and Ditlevsen (2021)*

    **Author's reply:**
    -

    **Changes in manuscript:**
    Suggestion followed.

14. *Line 345: space after (Eq. A2)*

    **Author's reply:**
    -

**Changes in manuscript:**
Suggestion followed.

15. *Table B1 caption: "based on Cimatoribus et al. (2014)". similar in B2-B4.*

   **Author's reply:**
   -

   **Changes in manuscript:**
   Suggestion followed.

**References**

- Ridgwell, A., Zondervan, I., Hargreaves, J. C., Bijma, J., and Lenton, T. M.: Assessing the potential long-term increase of oceanic fossil fuel CO2 uptake due to CO2-calcification feedback, Biogeosciences, 4, 481–492, https://doi.org/10.5194/bg-4-481-2007, 2007.

- O'Neill, C. M., Hogg, A. McC., Ellwood, M. J., Eggins, S. M., and Opdyke, B. N.: The [simple carbon project] model v1.0, Geosci. Model Dev., 12, 1541–1572, https://doi.org/10.5194/gmd-12-1541-2019, 2019.

- Toggweiler, J. R.: Origin of the 100,000-yr time scale in Antarctic temperatures and atmospheric CO2, Paleoceanography, 23, PA2211, https://doi.org/10.1029/2006PA001405, 2008

- Bergman, N.M., Lenton, T.M., Watson, A.J., 2004. COPSE: A new model of biogeochemical cycling over Phanerozoic time. American Journal of Science 304, 397–437

- Royer, D.L., 2014. Atmospheric CO2 and O2 during the Phanerozoic: tools. Patterns, and Impacts 251–267

- Zeebe, R. E.: LOSCAR: Long-term Ocean-atmosphere-Sediment CArbon cycle Reservoir Model v2.0.4, Geosci. Model Dev., 5, 149–166, https://doi.org/10.5194/gmd-5-149-2012, 2012.

---

## Author Comment (AC2)

**MS-No.:** ESD-2023-30

**Title:** Potential effect of the marine carbon cycle on the multiple equilibria window of the Atlantic Meridional Overturning Circulation

**Authors:** Amber A. Boot, Anna S. von der Heydt and Henk A. Dijkstra

**Point-by-point reply to reviewer #2**

June 7, 2024

We thank the reviewer for their careful reading and for the useful comments on the manuscript.

*In this manuscript, Boot and co-authors couple a physical box model of the AMOC to a carbon cycle box model. This tackles an interesting and largely unanswered question of how the carbon cycle and the AMOC influence each other, with a particular focus on whether and how the carbon cycle may impact the stability of the AMOC. The paper is overall well written. It builds on previous work such that the two box models are well established in their own right. I have two (related) main concerns and in the current form of the manuscript I was not able to tell whether these concerns are indeed pointing to fundamental issues with the approach and results, or whether it is rather an issue of presentation.*

*1) Is the coupling of the 2 full models needed to answer what I interpret as the central question: how does the MEW depend on atmospheric CO2 concentrations? This is related to another fundamental aspect I am concerned with: The very purpose of idealized box models is to reduce the complexity of a system to a small number of leading-order processes which can then be probed in detail to gain intuitive understanding. The model developed here with  30 ODEs is so complex that I wonder whether much intuitive understanding can be gained? Furthermore, from the figures presented it appears that many of the processes included have no or barely any notable impact on the processes that are being studied (see the overlapping curves in Fig 3 and the many almost identical lines in Figure 5). From my reading of Figure 4 a key process driving changes in MEW is the increase of Es with increased CO2? In that case, why not, for example, take the physical AMOC model and force it with Es (as constrained by the CMIP6-derived CO2) and consider the resulting changes in the MEW? Although I wonder whether this would be*

*rather similar to the original work of Cimatoribus et al (2014)?*

**Author's reply:**
The main reason for coupling the two models is to study whether feedbacks in the carbon cycle have a major influence on AMOC dynamics in steady state. The main idea here is that the carbon cycle responds to changes in the AMOC, resulting in a response in atmospheric $pCO_2$. This influences the atmosphere and therefore can influence the AMOC. By just forcing the model with $E_s$ we would not be able to capture the feedbacks in the carbon cycle, and the feedbacks between the AMOC and the carbon cycle and would indeed be similar to the sensitivity studies in Cimatoribus et al. (2014). We therefore believe that coupling the two models is essential for the overarching research questions of this work.

The total size of the system (i.e. 30 ODEs) is indeed large, but this is mainly because the carbon cycle is in itself very complex. To be able to capture carbon cycle dynamics we need 3 state variables per box (dissolved inorganic carbon, total alkalinity and a nutrient). This is the main reason for the relatively large problem size. Intuitively understanding the results is difficult, but this is inherent to studying carbon cycle dynamics, because it is such a complex system where biological, chemical and physical processes are intertwined. However, the most important carbon cycle variable for this study is atmospheric $pCO_2$. Changes in atmospheric $pCO_2$ can more easily be understood because it indirectly depends on the amount of carbon burial in the sediments. The burial rate is dependent on biological production and dissolution of calcium carbonate ($CaCO_3$) in the water column. To understand these processes, we do not need to have a full understanding of all the carbon cycle variables, which makes understanding the results already much simpler.

We have included the different feedbacks to see whether non-linear carbon cycle feedbacks can have a major influence of the multiple equilibria window (MEW) of the AMOC. What Fig. 3 shows is indeed that the effects of most feedbacks on AMOC dynamics are typically small when simulated under the same amount of carbon. We already made a selection to only highlight the feedbacks that are important for the conclusions in the main text. The BIO feedback is included because without it we cannot simulate an off branch. The $E_s$ feedback is included to couple the physical climate to the carbon

cycle. The FCA feedback is included because it changes the carbon cycle dynamics as seen in Fig. 3b, and especially when run under different carbon contents (Fig. 5). Since we are also using experiments with different amounts of carbon, and $CO_2$ concentrations, it was essential to also include the effects of temperature. We opted to use a low climate sensitivity ($CS_{Lo}$) and a high climate sensitivity ($CS_{Hi}$) to capture uncertainty in the climate sensitivity and to more clearly show the effect of the temperature feedback on the results.

**Changes in manuscript:**
No changes necessary.

*2) Is the combined model suitable to probe the size of the MEW? In my reading of the results, the size of the MEW (the distance between the dot-dashed and dashed lines in Fig 3) is barely impacted at all by accounting for different processes - even when the CO2 concentrations (right column of Fig 3) change quite notably. Similarly, the MEW size in Fig 5 is either completely or mostly insensitive to changes in the processes that are accounted for and also to total carbon content. I find this quite remarkable, since this is a very complex non-linear model and the authors consider a wide range of feedbacks and forcings etc, yet the MEW is largely constant. Again, as far as I can tell the main sensitivity is to Es (or atmospheric CO2) in Fig 4. This makes me wonder whether the title of the study should rather be something along the lines of "Robustness of AMOC MEW to changes in marine carbon cycle"?*

**Author's reply:**
The main response is indeed due to the sensitivity of the model to $E_s$ as presented in Fig. 4 and the sensitivity of $E_s$ to atmospheric $pCO_2$. We disagree with the reviewer that the MEW is mostly insensitive to total carbon content. In Fig. 5a for the bottom three cases (i.e. FCA, $CS_{LO}$, $CS_{HI}$) the MEW increases by approximately 20% as total carbon content increases by about 50%. We do show that the inclusion of certain non-linear carbon cycle feedbacks does not alter the response of the MEW. In that sense the suggested title would fit maybe better to the manuscript. However, since the MEW is, in our opinion, quite sensitive to the amount of carbon in the system, we do not think the suggested title adequately captures the conclusions of this paper.

**Changes in manuscript:**
No changes necessary.

*These comments are intended to highlight the questions that arose for me when I read the manuscript, and as I said above much of my skepticism may be the result of a lack of clarity of presentation. The other reviewer had some constructive ideas of how the presentation could be improved and that may alleviate some of my concerns above as well. I will further add that I have little expertise in the carbon cycle aspect of this work, which certainly hindered my interpretation. Nevertheless, I believe that a substantial reduction in the complexity of the model and the range of feedbacks and other processes may be required to be able to meaningfully shed light on the governing processes. As it stands, I found it difficult to assess the value of both the approach and the results.*

**Author's reply:**
Reviewer 1 indeed had some helpful suggestions on the presentation that we will follow. This will help to make the paper much clearer. As explained under comment 1, the carbon cycle is inherently complex, so we cannot reduce the complexity of the model much further. However, we will clarify the role of the additional feedbacks following suggestions of reviewer 1.

**Changes in manuscript:**
No additional changes necessary.

*As a final comment, I was noting the absence of any model validation or comparison to previous formulations. At one point the authors state that they had to add two boxes to ensure realistic $CO_2$ values. The original version apparently had very low $CO_2$ under AMOC collapse, and the authors state that most previous modeling studies found increases in $CO_2$ under AMOC collapse. However, the results in Fig. 3 still show substantial reductions in $CO_2$ when going from the AMOC "on" to the "off" state. In my reading this prompts open questions as to how this work compares to previous studies. To instill confidence in this novel coupled model, I would argue that some form of validation is needed.*

**Author's reply:**
With the used solution method, we can only solve for steady state solutions.

This makes it difficult to compare our results to other studies since these generally use time dependent simulations (see e.g. Gottschalk et al., 2019). Since we study the steady state response we do not expect that our model shows the same response as in these studies. However, our results have a similar order of magnitude as the studies which gives us confidence that the model is valid for our application.

**Changes in manuscript:**
We will mention how our results compare to the transient studies in Gottschalk et al. (2019).

**References**

- Gottschalk, J., Battaglia, G., Fischer, H., Frölicher, T. L., Jaccard, S. L., Jeltsch-Thömmes, A., Joos, F., Köhler, P., Meissner, K. J., Menviel, L., Nehrbass-Ahles, C., Schmitt, J., Schmittner, A., Skinner, L. C., and Stocker, T. F.: Mechanisms of millennial-scale atmospheric CO2 change in numerical model simulations, Quaternary Science Reviews, 220, 30–74, https://doi.org/https://doi.org/10.1016/j.quascirev.2019.05.013, 2019.

---

## Referee Report (RR1)

Review of Boot et al.: "Potential effect of the marine carbon cycle on the multiple equilibria window of the Atlantic Meridional Overturning Circulation"

This paper studies the width of the range of existence of multiple stable equilibria of the AMOC, as a function of different coupling mechanisms between a box-model for ocean circulation and carbon-cycle model studying the relevant ODEs with the continuation software AUTO. Both models have been previously studied and validated in a range of publications. The methods and the results are innovative and relevant for understanding AMOC stability and tipping in a range of climate states, they allow to identify and discuss relevant mechanisms and they certainly demonstrate that, in principle, a feedback between AMOC and carbon cycle is possible. Still, additional work is required, in my opinion, to organize in a clearer and more logical way the presentation of both the methods and the results.

In particular:

- The AMOC box model is presented in section 2.1 with 5 boxes but these are then extended to 7 (with addition of two boxes for the Indo- Pacific) in section 2.3.

- The coupled model is further modified when discussing the solution method in section 2.4, dropping the deep Atlantic box and substituting it with a global conservation constraint. This further change to the model structure occurs after having already discussed various additional coupling mechanisms in section 2.3 (which instead are fundamental for the further discussion of the results in section 3). An additional observation is that while the individual components (AMOC box model and CC model) have been previously applied in the literature, with these modifications we are now talking about quite different models, to what extent are these now comparable to the 'full' version of the components ?

- The different couplings (identified with BIO, $E_s$, FCA, $CS_{LO}$, $CS_{HI}$ by the authors) are mostly introduced in section 2, but FCA is described later, in the results section 3.1 instead. In general it would be good to introduce these labels close to the equations or maybe add equation numbers in table 1.

- The couplings presented in the main text (additional ones are introduced in the appendix) appear a bit like mixed bag of random choices. Some additional discussion on why these should be considered important and relevant couplings (also with reference to the literature) would be recommended. Which feedbacks could be expected associated with these couplings?

- When the SST dependence on atmospheric CO2 concentrations is introduced, with a simple model of climate sensitivity, it would be good to remind the reader what role SST plays in the model equations, which processes it does control.

- Actually a similar observation is valid also for other couplings: the rain ratio coupling in eq. 4) could be accompanied by a short reminder on its role in the biogeochemical cycle (or at least just repeat the description from line 83)

- L217-210: It is said that when the BIO coupling is not used, then $PO_4$ concentrations become negative in the surface ocean under a collapsed AMOC regime. This is not shown in any plot (not even in the appendix) and in general this sounds quite ominous: negative concentrations? Is this a numerical issue? Which mechanism leads to the drop in concentrations if a fixed biological export production in the surface boxes is used. If this is so, why is the BIO coupling actually considered an option and not integrated permanently in the model?
- Table 1 is introduced in line 237, after already on the previous page the impact of different couplings has been discussed.

Other questions are:

- To what extent are these results sensitive to particular modelling choices (such as for example the depth of the boxes?)

- The relationship between $E_s$ and atmospheric CO2 concentrations derived from CMIP6 models and described in eq. 8 could also be interpreted as a function of temperature, so should its use not be linked also to the activation of the climate sensitivity feedback?

- L228 and onwards: the total carbon content in the ocean+atmosphere system is kept fixed. I believe that some additional explanation on this hypothesis is due. What about carbon in terrestrial vegetation and soil carbon ?

- L237 onwards: there is no reference to the effect of introducing the $E_s$ coupling compared to the BIO case alone.

- Why were only these 'incremental' combinations of couplings explored? Is there a reason why only certain combinations should be used? I understand that considering all combinations might be confusing but maybe a short comment would be good.

-  L239: I might have missed it, but is there an explanation why the FCA coupling increases atmospheric CO2 ?

- The fact mentioned on line 255 that the on-branch becomes unstable before reaching the saddle-node bifurcation (due to a Hopf bifurcation):
  1) is this an hypothesis or was it confirmed by AUTO?
  2) Please clarify somewhere if the MEW is defined as the range between the saddle-node bifurcations or between the left saddle-node of the off-branch and the hopf bifurcation on the on-branch.

- The top axis of Figure 4 reports CO2 values between about 50 and 750 with $E_s$ varying between 0.25 and 0.50. Compared with fig 2c this does not look like the same fit (in that figure CO2 between 400 and 1200 has $E_s$ between 0.4 and 0.5).

- The fact that changes in SST (as modelled by eq. 3) do not affect ocean circulation in the model is discussed in the conclusions but indeed this might be a major drawback. Particularly through arctic amplification feedbacks, changes in the mean state can be associated with important changes in the meridional gradient of temperatures. Maybe the discussion on this point in the conclusions could be expanded.

- I realize that this is outside the scope of this study, but processes linked to sea-ice represent a major element affecting the strength of the AMOC, a comment might be in order. This could also be linked with the missing dependence of AMOC on model temperature which the authors recognize in the discussion.

- In the conclusions it could be beneficial to add a short comparison of these results (in particular the identification of the most important mechanisms) with other studies, maybe based on proxy data or on modelling with more complex climate models.

Minor issues:

- Line 177: for reproducibility it would be better to list somewhere which 28 CMIP6 models were used, which ensemble members
- Line 301: "These clear and plausible mechanisms…." → Which mechanisms? The previous sentence is about the CMIP6 fit ….. probably this paragraph (or the previous one) is not in the right place.

---

## Author Response (AR2)

**MS-No.:** ESD-2023-30

**Title:** Potential effect of the marine carbon cycle on the multiple equilibria window of the Atlantic Meridional Overturning Circulation

**Authors:** Amber A. Boot, Anna S. von der Heydt and Henk A. Dijkstra

**Point-by-point reply to reviewer #1 of the revision**

October 3, 2024

We thank the reviewer for their careful reading and for the useful comments on the manuscript.

*This paper studies the width of the range of existence of multiple stable equilibria of the AMOC, as a function of different coupling mechanisms between a box-model for ocean circulation and carbon-cycle model studying the relevant ODEs with the continuation software AUTO. Both models have been previously studied and validated in a range of publications. The methods and the results are innovative and relevant for understanding AMOC stability and tipping in a range of climate states, they allow to identify and discuss relevant mechanisms and they certainly demonstrate that, in principle, a feedback between AMOC and carbon cycle is possible. Still, additional work is required, in my opinion, to organize in a clearer and more logical way the presentation of both the methods and the results.*

*In particular:*

1. *The AMOC box model is presented in section 2.1 with 5 boxes but these are then extended to 7 (with addition of two boxes for the Indo- Pacific) in section 2.3.*

   *The coupled model is further modified when discussing the solution method in section 2.4, dropping the deep Atlantic box and substituting it with a global conservation constraint. This further change to the model structure occurs after having already discussed various additional coupling mechanisms in section 2.3 (which instead are fundamental for the further discussion of the results in section 3). An additional observation is that while the individual components (AMOC box model and*

*CC model) have been previously applied in the literature, with these modifications we are now talking about quite different models, to what extent are these now comparable to the 'full' version of the components ?*

**Author's reply:**
The AMOC dynamics are still very comparable to the literature (i.e. Cimatoribus et al., 2014; Castellana et al., 2019). We had to retune the model to keep $CO_2$ concentrations similar to the ones of the SCP-M. The different ocean circulation and box structure does change the model quite a bit. However, the most important aspects of the model are the carbon cycle dynamics. In the uncoupled case, these are still exactly the same. When couplings are introduced, we obviously change the model, the effects of these changes are one of the aspects we investigate in this study.

**Changes in manuscript:**
We have added a remark of the connection between the models in revised Section 2.3.

2. *The different couplings (identified with BIO, $E_s$ , FCA, $CS_{LO}$, $CS_{HI}$ by the authors) are mostly introduced in section 2, but FCA is described later, in the results section 3.1 instead. In general it would be good to introduce these labels close to the equations or maybe add equation numbers in table 1.*

   **Author's reply:**
   Suggestion followed.

   **Changes in manuscript:**
   We have made sure that the labels are introduced near the equations, and we have included equation numbers in table 1.

3. *The couplings presented in the main text (additional ones are introduced in the appendix) appear a bit like mixed bag of random choices.*

*Some additional discussion on why these should be considered important and relevant couplings (also with reference to the literature) would be recommended. Which feedbacks could be expected associated with these couplings?*

**Author's reply:**
Suggestion followed.

**Changes in manuscript:**
We have added additional motivation on the choices of the couplings. We also added an extra explanation on how a coupling would change the dynamics of the system.

4. *When the SST dependence on atmospheric $CO_2$ concentrations is introduced, with a simple model of climate sensitivity, it would be good to remind the reader what role SST plays in the model equations, which processes it does control.*

**Author's reply:**
Suggestion followed.

**Changes in manuscript:**
We have included a short explanation in the revised text.

5. *Actually a similar observation is valid also for other couplings: the rain ratio coupling in eq. 4) could be accompanied by a short reminder on its role in the biogeochemical cycle (or at least just repeat the description from line 83)*

**Author's reply:**
Suggestion followed.

**Changes in manuscript:**
We have included a short explanation in the revised text.

6. *L217-210: It is said that when the BIO coupling is not used, then $PO_4$ concentrations become negative in the surface ocean under a collapsed AMOC regime. This is not shown in any plot (not even in the appendix) and in general this sounds quite ominous: negative concentrations? Is this a numerical issue? Which mechanism leads to the drop in concentrations if a fixed biological export production in the surface boxes is used. If this is so, why is the BIO coupling actually considered an option and not integrated permanently in the model?*

   **Author's reply:**
   In the original SCP-M, biological export production is constant. In our coupled model, as the AMOC collapses, advection of $PO_4$ into box n decreases a lot. At some point on the unstable branch, the source term of $PO_4$, i.e. mixing of $PO_4$ into box n through $r_N$ and the AMOC, becomes smaller than the constant export production due to a weak AMOC and $PO_4$ concentrations will become negative. This shows that the original SCP-M, and our model without the BIO coupling, are not able to accurately simulate the carbon cycle of an AMOC off state because of missing processes. A main missing process is that biological production will decrease if nutrient concentrations decrease because of increased nutrient limitation. This process is captured in the BIO coupling which enables the model also to simulate a reasonable carbon cycle at the AMOC off state. It is a good suggestion to integrate the coupling permanently in the model, and if this model will be used in further research, that will also be the case. However, here we introduce the coupled model, i.e. the coupling between the carbon cycle processes of the SCP-M and with the ocean dynamics of the Cimatoribus box model, and we wanted to start off with staying as close to both original models as possible and then add new processes. This means we wanted to start of with a constant export production as modelled in the SCP-M and investigate how adding additional process affects the model.

   **Changes in manuscript:**
   We have clarified the text around L217 to better explain the reason for the negative $PO_4$ concentrations.

7. *Table 1 is introduced in line 237, after already on the previous page the impact of different couplings has been discussed.*

   **Author's reply:**
   It would indeed be more convenient if the table is placed earlier in the text.

   **Changes in manuscript:**
   The table is placed earlier in the revised text.

*Other questions are:*

1. *To what extent are these results sensitive to particular modelling choices (such as for example the depth of the boxes?)*

   **Author's reply:**
   We have tested the sensitivity to several variables, among which the rain ratio, depth of box n, and the strength of the global overturning circulation ($\psi_1$). Both the depth of box n and the global overturning circulation had little effect on the results. The rain ratio does have a strong effect on the $CO_2$ concentrations because it plays an important role in the burial of carbon in the sediments. The value of the rain ratio was chosen such that the atmospheric $CO_2$ concentrations on the AMOC on branch are around pre-industrial concentrations.

   **Changes in manuscript:**
   No changes necessary.

2. *The relationship between $E_s$ and atmospheric $CO_2$ concentrations derived from CMIP6 models and described in eq. 8 could also be interpreted as a function of temperature, so should its use not be linked also to the activation of the climate sensitivity feedback?*

**Author's reply:**
It is indeed possible that the increased freshwater flux is also related to the temperature. It would therefore make sense to link it to the activation by the climate sensitivity feedback. However, by decoupling them as we did in this study, we can study the separate effects of both the change in temperature and the change in salinity. This is especially convenient because in this model set up the change in temperature mostly affects the carbon cycle dynamics while the salinity changes mostly affect the AMOC dynamics. However, we believe it is good to discuss this choice more explicitly in the text.

**Changes in manuscript:**
We have included a few sentences on this choice in the revision.

3. *L228 and onwards: the total carbon content in the ocean+atmosphere system is kept fixed. I believe that some additional explanation on this hypothesis is due. What about carbon in terrestrial vegetation and soil carbon ?*

**Author's reply:**
In this model we do not consider the terrestrial biosphere and therefore do not take changes in vegetation and soil carbon into account.

**Changes in manuscript:**
The model detail has been clarified.

4. *L237 onwards: there is no reference to the effect of introducing the $E_s$ coupling compared to the BIO case alone.*

**Author's reply:**
Correct. We had not included one because there were hardly any changes in the model. However, for completeness, we have added it now.

**Changes in manuscript:**
A sentence discussing the $E_s$ coupling compared to the BIO case has been added.

5. *Why were only these 'incremental' combinations of couplings explored? Is there a reason why only certain combinations should be used? I understand that considering all combinations might be confusing but maybe a short comment would be good.*

   **Author's reply:**
   In principle, we could have performed many other combinations of the feedbacks. However, this would make it quite complicated to describe everything clearly in the manuscript, and indeed would probably lead to confusion. This is why we try to keep the experiments presented in the main text relatively simple and use these incremental steps to keep all the different cases relatively similar in set up. The exact choices presented in the main text are based on what feedbacks had the most pronounced effect on the AMOC and the MEW.

   **Changes in manuscript:**
   We have added a comment one the motivation of the used cases.

6. *L239: I might have missed it, but is there an explanation why the FCA coupling increases atmospheric $CO_2$ ?*

   **Author's reply:**
   This is indeed not mention explicitly in the main text. The FCA coupling adjusts the rain ratio, i.e. the relative amount of $CaCO_3$ in the biological export production, which affects the amount of DIC and Alk burial in the sediments. In the setting used here, the FCA coupling reduces the rain ratio, lowering the DIC and Alk burial in the sediments. As the river influx needs to balance the burial in the sediments, $CO_2$ concentrations decrease to lower the river influx.

   **Changes in manuscript:**

An explanation has been added in the revised paper.

7. *The fact mentioned on line 255 that the on-branch becomes unstable before reaching the saddle-node bifurcation (due to a Hopf bifurcation): 1) is this an hypothesis or was it confirmed by AUTO? 2) Please clarify somewhere if the MEW is defined as the range between the saddle-node bifurcations or between the left saddle-node of the off-branch and the hopf bifurcation on the on-branch.*

   **Author's reply:**
   1) This is confirmed by AUTO, and 2) we have defined the MEW as the range between the two saddle-node bifurcations.

   **Changes in manuscript:**
   This has been clarified in the revision.

8. *The top axis of Figure 4 reports $CO_2$ values between about 50 and 750 with $E_s$ varying between 0.25 and 0.50. Compared with fig 2c this does not look like the same fit (in that figure $CO_2$ between 400 and 1200 has $E_s$ between 0.4 and 0.5).*

   **Author's reply:**
   Correct; thanks.

   **Changes in manuscript:**
   Figure 4 has been corrected.

9. *The fact that changes in SST (as modelled by eq. 3) do not affect ocean circulation in the model is discussed in the conclusions but indeed this might be a major drawback. Particularly through arctic amplification feedbacks, changes in the mean state can be associated with important changes in the meridional gradient of temperatures. Maybe the discussion on this point in the conclusions could be expanded.*

**Author's reply:**
We agree.

**Changes in manuscript:**
The discussion on this has been expanded in the revised paper.

10. *I realize that this is outside the scope of this study, but processes linked to sea-ice represent a major element affecting the strength of the AMOC, a comment might be in order. This could also be linked with the missing dependence of AMOC on model temperature which the authors recognize in the discussion.*

    **Author's reply:**
    We agree.

    **Changes in manuscript:**
    The discussion on this has been expanded in the revised paper.

11. *In the conclusions it could be beneficial to add a short comparison of these results (in particular the identification of the most important mechanisms) with other studies, maybe based on proxy data or on modelling with more complex climate models.*

    **Author's reply:**
    Suggestion followed.

    **Changes in manuscript:**
    We have included extra discussion, also based on comments from reviewer 2, where we compare our results to other studies.

*Minor issues:*

1. *Line 177: for reproducibility it would be better to list somewhere which 28 CMIP6 models were used, which ensemble members*

   **Author's reply:**
   We agree. A list of the models is included in the repository corresponding to the paper. However, we had already prepared a list to also include in the supplementary material, but for some reason it was not included in the submitted manuscript.

   **Changes in manuscript:**
   A list of models and what ensemble members are used have been included in the supplementary material.

2. *Line 301: "These clear and plausible mechanisms...." → Which mechanisms? The previous sentence is about the CMIP6 fit ..... probably this paragraph (or the previous one) is not in the right place.*

   **Author's reply:**
   There was indeed a mistake in the order.

   **Changes in manuscript:**
   The order of the text has been revised.

**References**

- Castellana, D., Baars, S., Wubs, F. W., and Dijkstra, H. A.: Transition Probabilities of Noise-induced Transitions of the Atlantic Ocean Circulation, Scientific Reports, 9, 20 284, https://doi.org/10.1038/s41598-019-56435-6, 2019

- Cimatoribus, A. A., Drijfhout, S. S., and Dijkstra, H. A.: Meridional overturning circulation: stability and ocean feedbacks in a box model, Climate Dynamics, 42, 311–328, https://doi.org/10.1007/s00382-012-1576-9, 2014.

**MS-No.:** ESD-2023-30

**Title:** Potential effect of the marine carbon cycle on the multiple equilibria window of the Atlantic Meridional Overturning Circulation

**Authors:** Amber A. Boot, Anna S. von der Heydt and Henk A. Dijkstra

**Point-by-point reply to reviewer #2 of the revision**

**October 3, 2024**

We thank the reviewer for their careful reading and for the useful comments on the manuscript.

*This manuscript has gone through one review cycle. While I was not involved with the first review cycle, I have read the two review comments and the responses by the authors, and I believe the authors sincerely responded to the review comments. This is an ambitious work trying to evaluate the role of fully interactive carbon cycle to the AMOC multiple equilibrium window. There is high interest in the control of tipping points in the earth's climate system.*

*My main concern is about this manuscript is the mismatch in the timescale of global carbon cycle (including weathering and whole ocean inventory change) and AMOC (being a regional climate phenomenon). Re-organization of the global carbon cycle seems to involve much longer timescales than that of AMOC equilibria. In the discussion section, the authors stated that "Though not a limitation in the model, it is good to note that the range of timescales in the carbon cycle model is larger than in the circulation model, which does not affect our results but does affect the time dependent response of the system". I disagree with the above statement. The authors highlighted the importance of the balance between river input and sedimentation. For carbonate weathering system, its timescales are considered to be 10k-100k years, and the silicate weathering is on the order of 100k-1M years. These timescales are much longer than the timescale of AMOC variability O(1k year). On the timescales relevant to river input and sedimentation, there are other important changes in physical climate system that are not considered in this study, such as orbital parameters and the growth/decay of continental ice sheets.*

*For shorter timescale relevant to the AMOC variability, the internal re-distribution of carbon and alkalinity within the ocean can play more im-*

*portant roles. For example, the authors did not discuss the effect of changing ocean ventilation to the partitioning of carbon between ocean and atmosphere. There are studies making significant progress in understanding the important role played by AMOC (e.g. Goris et al., 2018; Katavouta et al, 2021; Zhang et al., 2024). The authors argue that the equilibrium solution is primarily controlled by the balance between river input and sedimentation for the set of processes represented in the model. In reality, there could be different processes dominating at different timescales. My suggestions to the authors are (1) to clarify what are the relevant timescales for this study and (2) to reference the work by others who examined the role of AMOC on the ocean carbon cycle over different timescales, and (3) discuss the main results in the context of the existing literature. In the previous works based on CMIP historical/scenario runs, the ocean carbon uptake in the subpolar North Atlantic decreases in the future climate with potential slowdown of AMOC. At the superficial level, this seems at odds against the lower atmospheric pCO2 in the off-state. Please explain how the transient and equilibrium solutions are different with respect to the AMOC's role.*

**Author's reply:**
The change in timescales does not matter for our results since we are looking at steady states. Variability on shorter timescales, such as adjustment of the ocean to AMOC tipping, does not play a role a role here. Steady states are determined through parameter continuation, and transient behavior is not considered. We agree that this needs clarification.

We agree with the suggestion to include a more thorough discussion on how this work relates to existing literature.

**Changes in manuscript:**
We have added a discussion on the steady state approach for clarification. We have, also based on comments by reviewer 1, included a discussion where we compare our results to existing literature where we also highlight the timescales involved in different studies.